# Room temperature 3D carbon microprinting

Fernand E. Torres-Davila[1,2], Katerina L. Chagoya[3], Emma E. Blanco[4], Saqib Shahzad[5], Lorianne R. Shultz-Johnson[4], Mirra Mogensen[1,4], Andre Gesquiere[1,4], Titel Jurca[1,4,6], Nabil Rochdi [7,8], Richard G. Blair [6,9] ✉ & Laurene Tetard [1,2] ✉

Manufacturing custom three-dimensional (3D) carbon functional materials is of utmost importance for applications ranging from electronics and energy devices to medicine, and beyond. In lieu of viable eco-friendly synthesis pathways, conventional methods of carbon growth involve energy-intensive processes with inherent limitations of substrate compatibility. The yearning to produce complex structures, with ultra-high aspect ratios, further impedes the quest for eco-friendly and scalable paths toward 3D carbon-based materials patterning. Here, we demonstrate a facile process for carbon 3D printing at room temperature, using low-power visible light and a metal-free catalyst. Within seconds to minutes, this one-step photocatalytic growth yields rod-shaped microstructures with aspect ratios up to ~500 and diameters below 10 µm. The approach enables the rapid patterning of centimeter-size arrays of rods with tunable height and pitch, and of custom complex 3D structures. The patterned structures exhibit appealing luminescence properties and ohmic behavior, with great potential for optoelectronics and sensing applications, including those interfacing with biological systems.

Demand is ever-increasing for the controlled growth of carbon-based materials[1], in part because of the unique physical properties they can contribute to flexible electronics, devices for human and plant health monitoring[2,3], neuroscience[4], and beyond[5,6]. Advances in additive manufacturing, by means of material elimination and material transfer methods, have facilitated the design of conductive freeform structures, at the cost of multiple steps and limited spatial resolution[7,8]. A more straightforward approach to produce freeform structures of conductive materials[9], which afford appealing mechanical and electrical properties, good stability, and bio-compatibility, is lacking. However, 3D manufacturing of carbon faces longstanding challenges. Carbon growth on polymeric or fabric substrates is hindered by the high temperatures required in most approaches[6]. While laser-induced graphene electrodes have been obtained by polymer irradiation[10,11], the formation of on-demand out-of-plane electrodes and more complex assemblies is far from being trivial. In fact, high aspect-ratio structures[12] with diameters ranging from a few hundred nanometers to 10 µm that can be patterned in large arrays with tunable pitch below 150 µm or in multi-branched assemblies are currently unattainable with traditional nanofabrication and 3D printing.

Reducing the environmental and energy footprints of carbon manufacturing processes is also important, given the considerable global market. However, carbon microstructures and nanostructures are commonly produced by pyrolyzing organic precursors of synthetic or biological origin[13,14]. Such an approach involves energy-intensive multiple-step processes, such as fiber spinning, stabilization, and carbonization at temperatures typically above 1200 °C[13]. Catalytic conversion of hydrocarbons into solid carbon is another approach often considered for the production of nanostructures[15]. It involves

[1]NanoScience Technology Center, University of Central Florida, Orlando, FL, USA. [2]Department of Physics, University of Central Florida, Orlando, FL, USA. [3]Department of Mechanical and Aerospace Engineering, University of Central Florida, Orlando, FL, USA. [4]Department of Chemistry, University of Central Florida, Orlando, FL, USA. [5]Department of Materials Science and Engineering, University of Central Florida, Orlando, FL, USA. [6]Renewable Energy and Chemical Transformations (REACT) Cluster, University of Central Florida, Orlando, FL, USA. [7]Laboratory of Innovative Materials, Energy and Sustainable Development (IMED-Lab), Cadi Ayyad University, Marrakesh, Morocco. [8]Department of Physics, Faculty of Sciences Semlalia, Cadi Ayyad University, Marrakesh, Morocco. [9]Florida Space Institute, University of Central Florida, Orlando, FL, USA. ✉e-mail: Richard.Blair@ucf.edu; Laurene.Tetard@ucf.edu

transition metal catalysts (Ni, Fe, or Co) and temperatures ranging from 400 to 1500 °C, or equivalent high-energy sources[15]. Apart from these, the synthesis of carbon microstructures by hydrocarbon dehydrogenation is possible using laser-assisted chemical vapor deposition (L-CVD)[16]. When combined with high temperatures, L-CVD allows direct cracking of chemical bonds in the precursor molecules using ultraviolet or visible laser excitation at powers typically exceeding 1 W[17]. Its implementation is hindered by the critical requirement of maintaining a stable temperature at the structure's growth front to sustain growth, which can only be achieved by displacing the laser focus in concert with the structure's growth rate[16,17]. Overall, developing more eco-friendly scalable strategies will bear far-reaching energy savings[18].

We present the highly controlled, low-power visible light-assisted growth of carbon microstructures on a wide range of substrates at room temperature. This technique enables the 3D carbon micropatterning of well-controlled arrays or 3D complex assemblies with appealing properties for electronics, optoelectronics, and sensing. The process is enabled by a metal-free catalyst, defect-engineered hexagonal boron nitride (dh-BN) and yields carbon structures with illumination-dependent features.

## Results

### Versatile photocatalytic carbon growth at room temperature

Recent studies have shown that tailoring the local electronic density in two-dimensional (2D) h-BN layers confers a tunable chemical reactivity to this otherwise wide bandgap ( ~ 6 eV) material[19–21]. This reactivity is promoted by energy states introduced in the bandgap by defects induced during mechanical milling, as corroborated by density functional theory (DFT)[19,22]. Figure 1a shows loose defect-laden h-BN (dh-BN) powder obtained by ball milling (see Methods and Supplementary Information) and placed in a custom reactor pressurized with a hydrocarbon gas without exposure to air (Supplementary Fig. 2). A low-power (10–100 mW) 532-nm-wavelength laser (i.e., photon energy ~2.3 eV) focused on the powder surface with a 10× objective is used to activate hydrocarbon dehydrogenation. A color change from white to black (Supplementary Fig. 3) attests to the ongoing reaction. Further inspection of the reacted region obtained by rastering light with 10 μm steps and a dwell time of 30 s at each point revealed an array of well-aligned rod-shaped microstructures with a pitch of 10 μm, as shown in Fig. 1b. The region between carbon rods also underwent a rapid reaction at the photocatalyst surface while the laser moved from one point to the next. The reacted powder without carbon rod growth

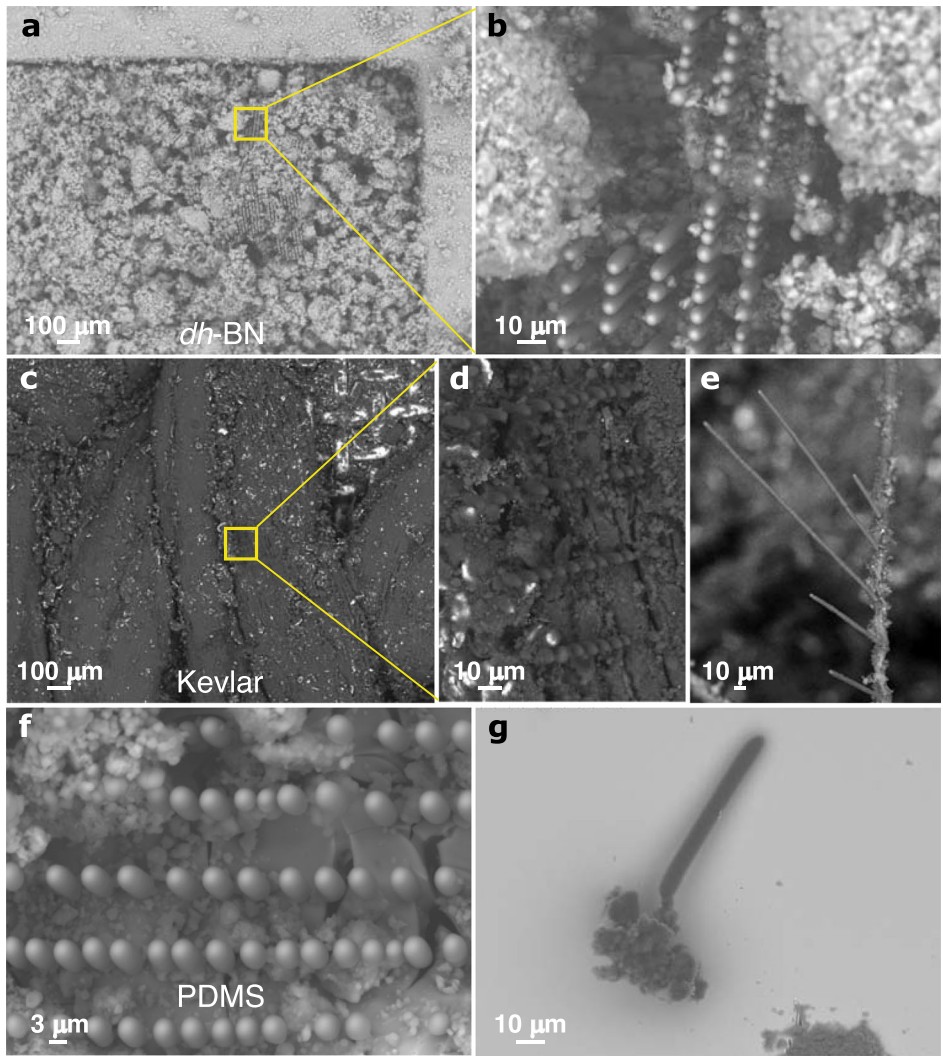

**Fig. 1 | Low-energy growth of carbon microrods on loose defect-engineered hexagonal boron nitride (dh-BN) powder. a** Scanning electron microscopy (SEM) image of the dh-BN powder placed in a well, including regions treated with laser light. **b** SEM image of the region marked in **a** revealing the presence of aligned rods in the powder. **c** SEM image of Kevlar coated with a processed thin layer of dh-BN powder. **d** SEM image of the region marked in **c** revealing the presence of aligned rods on the woven fibers of the textile. **e** SEM of carbon rods of various lengths grown on a single strand of Kevlar. **f** SEM of an array of carbon rods obtained on polydimethylsiloxane (PDMS). **g** SEM image of a single microrod extracted from the processed powder.

contains a carbon-rich matrix with encrusted *h*-BN clusters (brighter regions in Supplementary Fig. 4a).

Carbon rods were grown on textile and polymeric substrates using the same process. Figure 1c, d presents an array of carbon rods on Kevlar woven fibers while Fig. 1e shows multiple carbon rods produced with controlled pitch and lengths on a single strand of Kevlar fiber. The same approach was used to pattern carbon rods on polydimethylsiloxane (PDMS) (Fig. 1f), showing that the temperature of the process is low enough to overcome the current limitations in carbon growth processes, making it suitable for flexible electronics. Individual carbon rods isolated from the arrays reveal a growth seeded in the granular-shaped roots made of *dh*-BN surrounded by a carbon matrix. The rod diameters range from 2 to 10 μm depending on the growth conditions. Their length can be tuned from ~30 to 150 μm in the first few seconds of growth with an immobile objective. All rods exhibit smooth surfaces throughout the rod length (Fig. 1g).

## Morphology and chemical makeup of the carbon microstructures

During the growth of each microstructure, a strong luminescence is emitted from the illuminated region upon formation of the carbon motifs interacting with *dh*-BN defects, followed by the formation of graphitic carbon (Supplementary Figs. 7–9). Hence, the fluorescence emission provides an indicator to guide the microstructure growth progress (Supplementary Movie 1). Ex-situ Raman analysis of the microrods shows the presence of the graphitic (G) band at 1600 cm$^{-1}$, the defect (D) band centered at 1335 cm$^{-1}$, the G' band, and the D + G band in the 2500–3100 cm$^{-1}$ range. Cross-sections of microrods reveal a dense material with a core-shell structure (inset in Fig. 2a). Spectra collected on the cross-sectioned microrod indicate a significant difference in the graphitization level between the shell and the core (Fig. 2a). Graphitic domains about 3.1 nm in size were estimated in the shell compared to 5.4 nm in the core. On the other hand, the $sp^2$ phase organized in rings is predominant in the shell. X-ray diffraction (XRD) patterns acquired on an array of densely packed microrods confirm the graphitic nature of the structures (Fig. 2b), with a peak centered at 26.1°. The shift of this peak with respect to pure graphite ($2\theta = 26.4°$ according to the reference pattern JCPDF 41–1487) refers to a slightly higher average spacing between the $sp^2$ carbon layers (~0.342 nm) compared to that of graphite (~0.337 nm). This is attributed to the turbostratic nature of the material, which is supported by the Lorentzian fit of the G' peak centered at ~2680 cm$^{-1}$ (see Supplementary Fig. 7). The peak centered at 26.6° is attributed to *h*-BN (JCPDS Card no. 85-1068). The binding energies of the tips of the same microrod array (Fig. 2c) and the shells of individual microrods (Supplementary Figs. 14 and 15), studied by X-ray photoemission spectroscopy (XPS), further confirm the graphitic character of the microrods. The C 1*s* narrow scan on the microrod tips shows a prominent peak at 284.7 eV corresponding to C−C bonds, two peaks at 286.45 and 288.55 eV of oxidized carbon (C−O and C=O bonds), and two additional peaks at 283.45 and 287.65 eV. These features originate mainly from carbon pentagonal rings (and C−B bonds to a lesser extent) and from carbon in pyridone or pyrrolic nitrogen, respectively[23,24]. The B 1*s* and N 1*s* spectra exhibit features corresponding to *dh*-BN (Supplementary Fig. 11), while the O 1*s* spectrum depicts features of oxidized carbon. The valence band and C KLL Auger signals indicate that the microrod tips are made of $sp^2$-$sp^3$ hybridized carbon, with the valence band maximum located at about 0.82 eV below the Fermi level (Supplementary Figs. 12 and 13). Conversely, high resolution (HR) XPS analysis indicates that the microrods' shell is made of pure (boron- and nitrogen-free) graphitic structures with carbon defect sites (Supplementary Fig. 15), in agreement with the Raman spectroscopy results (Fig. 2a). The increase in graphitic order toward microrods' roots further supports the proposed growth model (Fig. 3g). The region immediately underneath the microrod tip, exposed by ionic beam sputtering,

exhibits a signal corresponding to a nitrogen-deficient CBN compound, likely with a prominent boronated graphite composition (Supplementary Fig. 12). We infer that the laser power, growth kinetics, and laser-induced heating mediate the rate of the graphitization process[16].

## Mechanisms of the photocatalytic growth

Arrays of identical 100-μm-long rods were printed with a pitch of 30 μm on a borosilicate substrate covered with a thin coating of *dh*-BN, as shown in the top (inset) and side views of Fig. 3a, illustrating the repeatability of the process. The pitch could be reduced to about 10 μm, which corresponds to the rods' diameter in this case, forming arrays of densely packed structures (Fig. 3b). Using the same approach, we examined the effect of process parameters on the microrods' growth and morphology. Photon energy above 2 eV is required to observe growth, which is consistent with the visible absorption of *dh*-BN (Supplementary Fig. 6). To the best of our knowledge, this constitutes the first demonstration of the visible photocatalytic activity of *dh*-BN[25]. Growth occurs predominantly in the longitudinal direction while the width remains constant for the investigated parameters (532-nm laser with a 10×, 20× or 50× objective) (Fig. 3c). When maintaining the laser focal plane fixed at the surface of the *dh*-BN coating, the growth rate exhibits a linear regime (of about 40 μm/s) for up to 1 s of illumination followed by a significant rate decrease (~0.1 μm/s after 30 s of illumination). Rods produced with illumination beyond 60 s with the immobile light exhibit identical lengths. Width variations could only be achieved by changing the profile of the focal volume (objective magnification, numerical aperture or other optical features) or the pressure of the reactant gas. Further investigations revealed that focusing the light at a magnification of at least 4× was required to observe carbon growth out of the plane for the parameters considered, even with more intense laser power (~300 mW). Interestingly, focusing the illumination up to 60 μm above the surface of the catalyst resulted in a ~1.5-fold increase in the rod length (Fig. 3d), while no rod was formed when the focal plane was at a distance exceeding 80 μm above the catalyst surface or 25 μm below the surface.

The total length of the structures can be significantly increased by refocusing the laser at the tip of the segment synthesized after each 30 s exposure (Fig. 3e and Supplementary Fig. 16), or by continuously moving the laser focus with the microstructure's tip (Fig. 3f and Supplementary Fig. 17). The lengths up to 2.99 mm and aspect ratios up to ~500 obtained with this process constitute a remarkable breakthrough, which is out of reach using traditional 3D printing[12] and other microscale fabrication processes. These findings suggest that the microrod tip is highly reactive. However, a vapor-liquid-solid mechanism[26], as observed during the growth of carbon nanotubes[27] and semiconductor nanostructures[28,29] is unlikely since metal catalysts require high temperatures for liquid metal supersaturation with dissolved atoms from the decomposed reactant gas that solidifies by segregation as temperature decreases. The growth mechanism presented here occurs at room temperature with a low power laser, thereby the non-metal catalyst is expected to remain far below the state of liquefaction, reported above 2900 °C for *h*-BN. In fact, temperatures above 900 °C may heal defects in *h*-BN, which would lead to a loss of activity toward carbon growth[30]. Experimental work carried out to date suggests a tip-growth mechanism with carbon compounds separated by photocatalytic dehydrogenation of the hydrocarbon molecules diffusing through the tip and binding to dangling bonds to form the rod (Fig. 3g).

## Functional properties of the carbon microstructures

The carbon structures present exciting opportunities for applications in optoelectronics and beyond. As shown in the photoluminescence (PL) intensity map of an individual rod (Fig. 4a), a more intense luminescence is emitted from the outer layer of the structure than from the

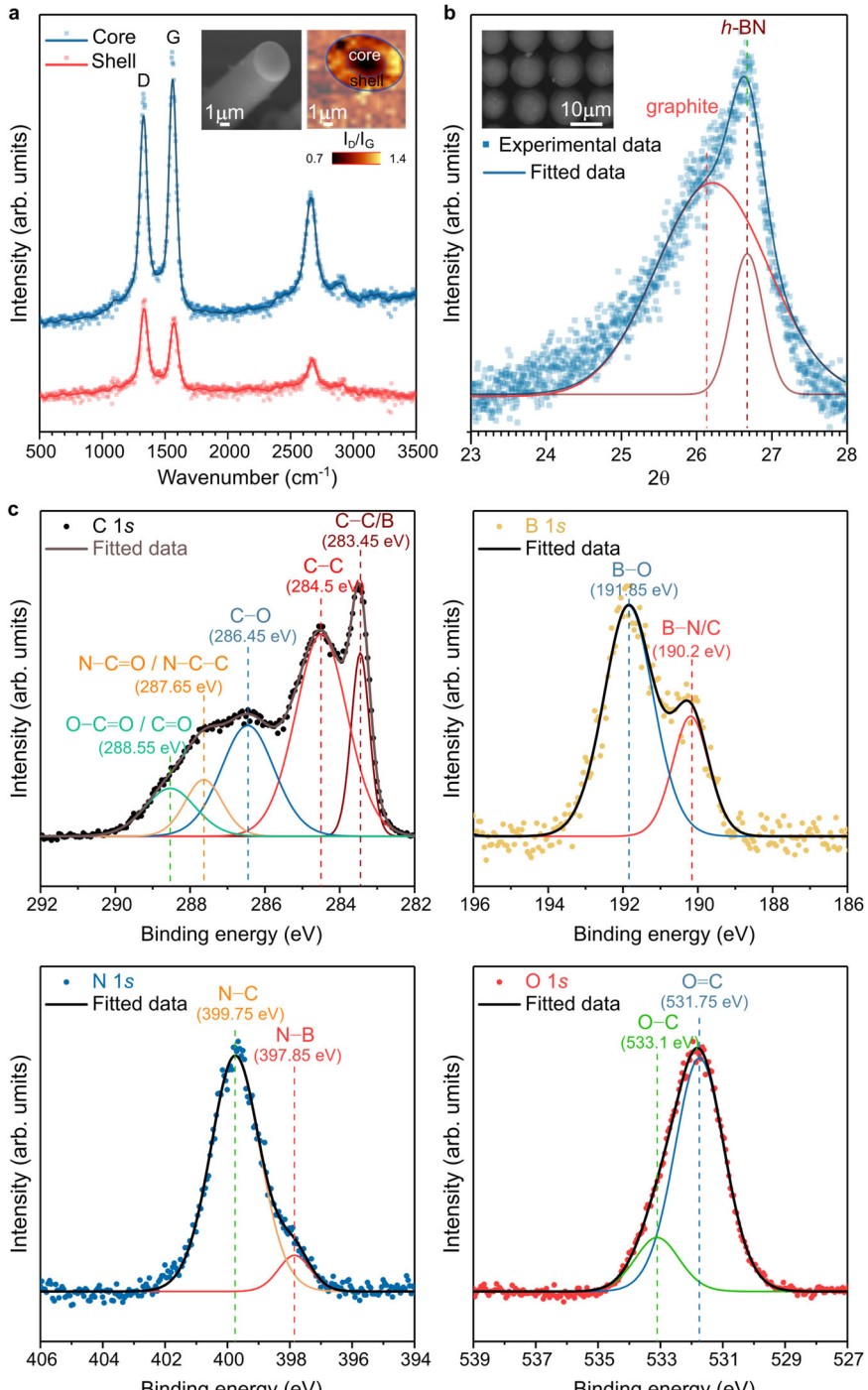

**Fig. 2 | Composition and structure of patterned carbon microstructures.**
**a** Raman spectra of the core (blue) and shell (red) obtained on a cross-sectioned carbon microstructure. Square symbols represent the experimental data, while solid lines represent the smoothed data. A SEM cross-sectional image with the corresponding Raman map of the $I_D/I_G$ variations is shown in inset. **b** XRD pattern (blue) of a dense array of microrods, as shown in the inset SEM image, with corresponding deconvoluted contributions of graphite (red) and $h$-BN (brown) **c** High

resolution (HR) XPS spectra of C 1$s$, B 1$s$, N 1$s$, and O 1$s$ core levels measured on the tips of 1.2 mm-long carbon microrod arrays patterned on $dh$-BN on a quartz substrate. The HR spectra (dot symbols) are decomposed into prominent Gaussian–Lorentzian features (colored lines labeled in each panel) with corresponding fitted curves (black solid lines) indicating the nature of chemical bonding. Source data are provided as a Source Data file.

core. Small clusters with very high PL intensity can be observed on the shell. PL spectra (Fig. 4b) reveal a broad emission in the visible range peaking between 540 and 566 nm. The emission from the core, shell, and cluster suggests slight differences in composition, in good agreement with the Raman maps of the structure's cross section (Fig. 2a). Variations in the PL spectra observed between the shell and

clusters can result from the formation of $sp^2$ or $sp^3$ structures during the reaction, as also evidenced by the Raman and XPS analyses. The reactivity of the shell was further confirmed by the ability to grow secondary branches originating from the shell of the first printed microrod (Fig. 3f). Growing multiple branches with different directions is possible from either the tip (Fig. 3e) or the shell (Fig. 3f), with an

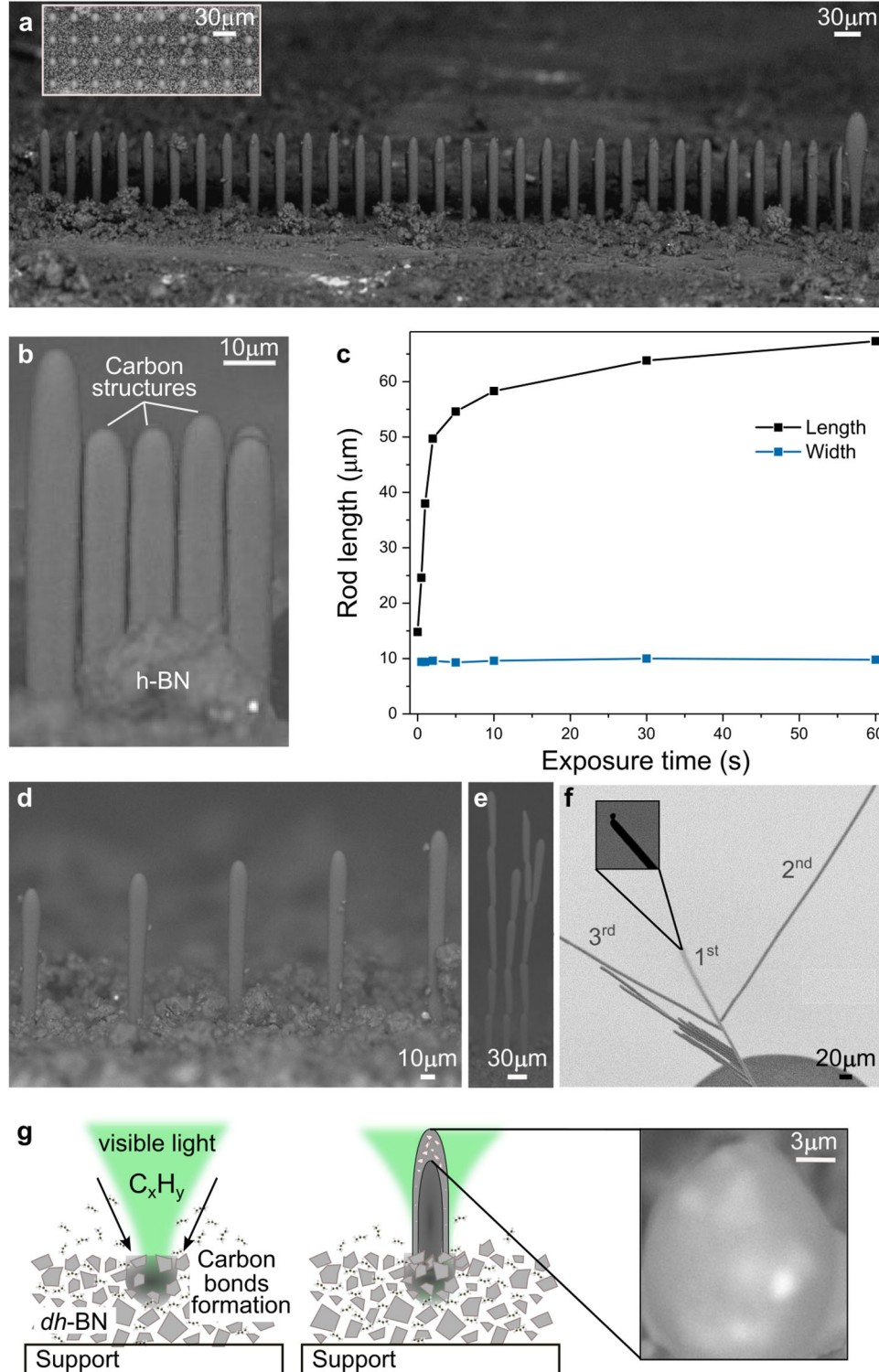

**Fig. 3 | Effect of growth conditions on the morphology of carbon microstructures and arrays. a** SEM image of 30-μm-pitch microrod array obtained upon 30 s exposure under 532 nm laser (100 mW) focused with a 20× objective on glass substrates coated with *dh*-BN, with top view of the array provided in the inset. **b** Dense microrod array obtained when reducing the pitch to the rod diameter. **c** Effect of exposure time on the length (black) and width (blue) of the carbon rods obtained with a 532 nm laser at 100 mW, with illumination focus fixed at the surface of the *dh*-BN coating. **d** Effect of changing the position of the laser focal plane from 20 μm (left rod) to 60 μm (right) above the surface. **e** SEM image of multi-branched microstructures obtained by successive segmented growth from the active tip. **f** SEM image of 3D multi-branched microstructures showing the activity of the shell. The length of each branch was obtained by moving the laser continuously during growth. **g** Schematics summarizing the growth steps of the carbon microstructures. The *dh*-BN powder is pressurized with a hydrocarbon gas. Upon exposure to focused visible light (532 nm, depicted in green), carbon growth is initiated, and the structure is formed following the illumination scheme. A SEM image of the carbon microrod tip is presented as inset. Source data are provided as a Source Data file.

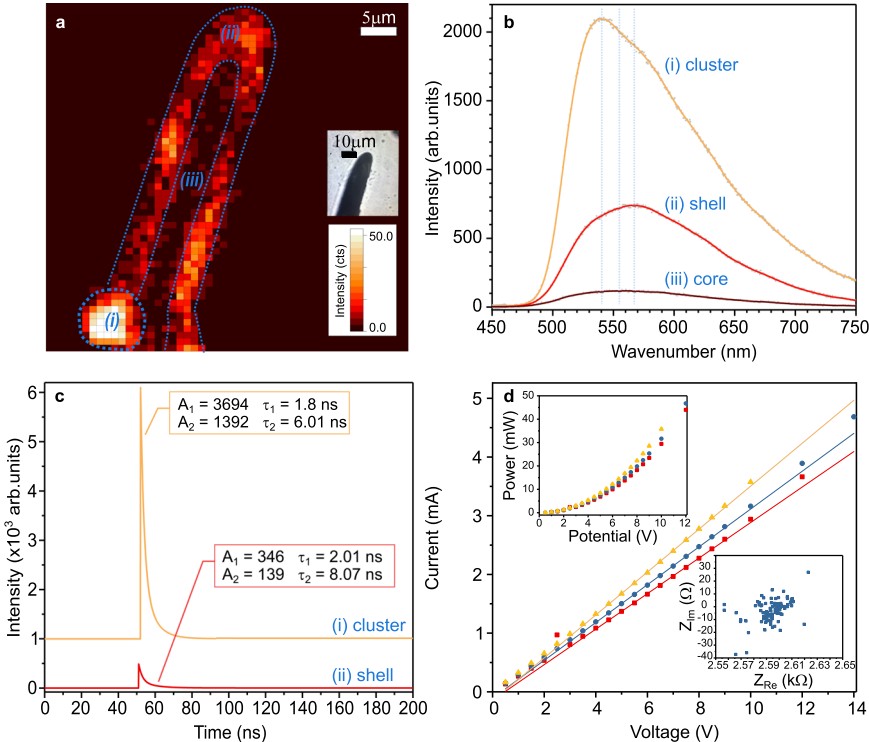

**Fig. 4 | Optical and electrical properties of the carbon microstructures.**
**a** Photoluminescence intensity map of a carbon rod displaying three regions with different intensities, a high intensity cluster (i) attached to the shell (ii) and the core (iii). An optical image of the rod is shown in the inset. **b** Corresponding photoluminescence spectra of the cluster (yellow), shell (red) and core (brown) of the microrod. Gray symbols represent the experimental data. Vertical dotted lines indicate the peak emission wavelengths. **c** Time-resolved photoluminescence

characteristics of the cluster (yellow) and the shell (red). **d** Current-voltage response of three 1.2 mm-long microstructures with a diameter of 2.54 μm (yellow, blue and red symbols and corresponding-colored lines representing the linear fit) with the corresponding power-voltage response in top left inset. The Nyquist plot of one microstructure is presented in the bottom right inset. Source data are provided as a Source Data file.

angle from a few to 90°. Continuous growth of nonlinear structures is also attainable. Hence, the room-temperature process is a unique approach for 3D micropatterning of carbon, circumventing the use of metals as catalysts and high temperatures for processing. As shown in Fig. 3f, 3D antennas similar to those observed on moths can be created in minutes. The nature of the reactive regions, with high PL, is further confirmed by excited state lifetime measurements (Fig. 4c). The time-resolved PL characteristics of the cluster and the shell exhibit a double-exponential behavior with a similar decay time $\tau_1$ of about 2 ns and a quite different decay time $\tau_2$ (of about 6 and 8 ns, respectively). The short-time component has previously been assigned to the emission dynamics of the zero-phonon line (ZPL) in carbon-enriched $h$-BN structures, while the long-time component was assigned to the phonon-assisted recombination[31]. The variations observed from pixel to pixel suggest that $dh$-BN is inhomogeneously distributed in the shell of the carbon structures in small quantities, with likely interactions of the carbon with $dh$-BN defects. Nonetheless, the carbon matrix is predominant in the structure as shown by Raman, XRD, and XPS measurements (Fig. 2). This provides an overall conductive material (Fig. 4d) suitable for 3D electronics. The I-V curves and Nyquist plot of the structures indicate that the long rods ( >1 mm) are simple resistors. Catastrophic failure was observed when an electrical power exceeding 40 mW was applied to 2.54-μm-diameter rods in air, over a length of 0.36 mm. This corresponds to a breakdown current density of $7.89 \times 10^5$ A/cm², which is better than copper, close to that of CVD graphene ($4 \times 10^7$ A/cm²)[32], but less than carbon nanotubes ($2.4 \times 10^9$ A/cm²)[33]. This remarkable finding could be transformative for interconnects and multi-level layouts in integrated circuits, in particular for flexible circuits.

Individual carbon microrods were also tested as strain and temperature sensors, as shown in Supplementary Figs. 19 and 20, respectively. Both devices exhibit high performance in terms of strain achieved and the fractional change in resistance resulting from the change in temperature (−35% between 20 and 100 °C)[34]. Flexible devices with the ability to monitor both temperature and strain fluctuations are very appealing for human health, while the combination of conductivity and fluorescence properties is particularly appealing for bioimaging and biosensing[35]. We show in Fig. 5 that the single microrod-based strain gauge sensor exhibits sensitivity about 30 times superior to a conventional gauge (gauge factor 2 μV/με at 4 V), as indicated by its gauge factor of −59.9 μV/με for a strain of 0.38%. This is notable as it is superior to gauge factors of 2-4 commonly reported for strain gauges[34]. Further deformation of the rod is possible thanks to its ultra-high aspect ratio, despite it being limited by the assembled sensor stack. Flexing a mounted microrod without the commercial strain gauge in the stack allowed for a strain of 0.55% to be applied without damaging the microrod (Supplementary Fig. 19). A loop (R - 243 μm) could be obtained when flexing an unmounted 5 μm-diameter microrod without breakage, which corresponds to a strain of ~ 2%. This strain is greater than the failure strain of some carbon fibers[36].

The discovery of this light-driven synthesis process introduces a facile, low-cost, and low-energy solution for freeform carbon 3D patterning. With the formation of carbon microstructures governed by the shape and orientation of the incident beam, shaping complex patterns of highly-ordered microstructures becomes attainable, with resolution outperforming traditional 3D printing. Ohmic behavior and luminescence features of the patterned microstructures pave the way to the formation of more intricate 3D conductive networks of

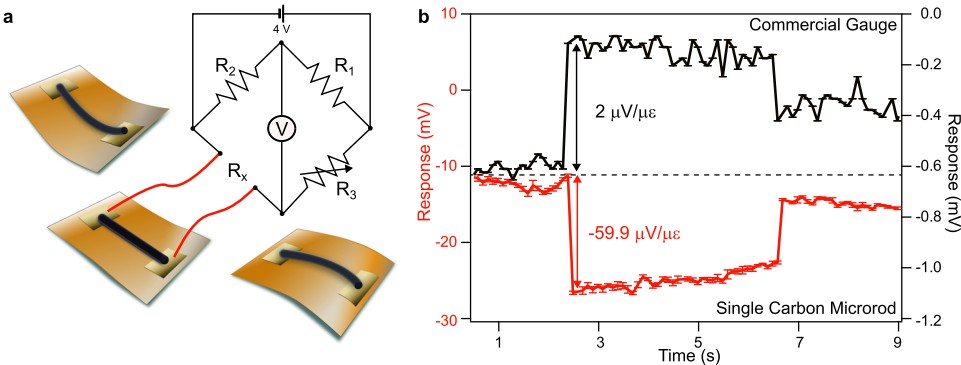

**Fig. 5 | Strain gauge single carbon microrod sensor. a** Schematics of the single carbon microrod-based strain gauge sensor using a Wheatstone bridge to achieve a precise resistance measurement of the carbon microrod undergoing deformation. **b** Response of the single carbon microrod-based strain gauge sensor (red curve) compared to a standard strain gauge sensor (black curve). Error bars represent the standard deviation of the measurements. Source data are provided as a Source Data file.

structures. The performance of single carbon microrod-based temperature and strain sensors were found to outperform conventional devices. As a result, it is expected that these materials will provide good interfaces with biological systems for sensing and stimulation purposes because of the absence of metallic contaminants responsible for cellular toxicity. Further beam shaping and environmental control provide additional ways to grow structures with tunable size, morphology, and composition. This perspective is further exciting if thought of in combination with defects engineering prospects for nanopatterning of reactive sites. In this respect, recent light-assisted local defects nanomachining in 2D materials[37] indicates that scalable manufacturing of reactive sites separated by nanoscale pitches and structured surfaces is within reach. These innovative micro- and nano-printing methods provide an eco-friendlier impetus to impact the next-generation of carbon materials for applications in flexible electronics, quantum information, sensing, catalysis, and beyond.

## Methods

### Catalyst engineering

Pristine *h*-BN powder (Saint-Gobain Ceramic Materials, Grade PCTF5) was dried in vacuo at 400 °C for 12 h and transferred to an argon-filled glove box for storage. Ball milling was performed by placing 2 g of *h*-BN into a 45 mL zirconia vial (SPEX 8005) while in the glove box, with one 19.05-mm-diameter zirconia ball ( ~ 26.6 g) and sealed with a silicone o-ring. The vial was placed in a SPEX Certiprep 8000 M mixer/mill, and the powder was processed for 120 min under inert conditions. The vial was transferred to an argon-filled glove box, where it was prepared on selected substrates before loading into the custom reactor cell.

### Carbon growth

The milled *dh*-BN powder was first loaded into the reactor, shown in Supplementary Fig. 2, under an inert environment. The cell was pressurized with the hydrocarbon gas (reactant gas, propene in this study) at a pressure of 40 psi (276 kPa). Pressure and temperature were monitored throughout the experiment. The operating temperature was 25 °C (referred to as room temperature hereafter).

Laser light was focused using a 10× objective (Zeiss, N.A. = 0.2) unless otherwise mentioned. The light was focused at or above the surface of *dh*-BN through a sapphire window. Laser power was varied from 10 to 100 mW for the measurements reported in this study. The light was initially focused on the powder for an extended period (from a few minutes to 3 h depending on the conditions of illumination) to initiate the first reaction site. The illumination duration and the pitch to create the arrays were controlled using the

hardware of a confocal Raman microscope, which provides fine control in X-Y and Z displacements, and an optical view with a notch filter (blocking the excitation line) to observe the reaction site with the camera.

### Material characterization

UV-Visible spectra of *h*-BN and *dh*-BN were acquired using an Agilent Cary 60 UV-Visible spectrophotometer. Thin films of the powders were prepared between two quartz glass slides, sealed to prevent exposure to air.

The reaction was monitored using Raman and PL spectroscopy. The measurements were performed on confocal optical microscopes (WITec Alpha 300RA for excitation at 532 nm and HORIBA LabRAM for excitation at 473, 633, 785, and 1064 nm). The excitation laser was maintained at a low laser power (below 10 mW) during Raman spectra acquisition to prevent activating any reaction. Rayleigh scattering was filtered using a notch filter before reaching the spectrometer. Using a grating (300, 600, or 1800 lines/mm), photons were collected with an integration time of 1 s (or other acquisition time otherwise specified) with the charge-coupled device (CCD) detector. 10 scans were averaged for each measurement unless otherwise indicated. Curve fitting to evaluate the spectral components was carried out with Fityk 1.3.1 after background removal of the constant noise.

Powder XRD of the carbon rods was performed using a PANalytical Empyrean powder X-ray diffractometer with a 1.8 kW copper source (Cu *Kα* = 1.5418 Å). Diffractograms were collected from 5 to 40 degrees (2*θ*) using 0.004-degree steps and 10.16 s of dwell time per step. The measurements were carried out on a sample encompassing a dense matrix for carbon rods printed over a 2 × 2 mm² region on a (511) silicon zero background plate. Analysis of the spectra was carried out following the Scherrer method[38] after extracting peak full width at half maximum (FWHM) by simultaneously fitting the (002) graphite peak and the narrower (002) *h*-BN peak.

Binding energies were studied using an X-ray photoelectron spectrometer (Thermo Scientific Escalab Xi + ) with an aluminum radiation source to generate a 1486.6 eV-energy K*α* excitation. The spot size was 200 µm, and the investigated energy range for survey scans spanned from 0 to 1350 eV. Charge compensation was used for all scans. Three scans were averaged for the survey spectra, with an energy step of 0.5 eV and a dwell time of 20 ms. The HR XPS spectra were recorded using an energy step of 0.05 eV and a dwell time of 50 ms. 10 or 25 scans were averaged for each element's narrow spectrum. The HR XPS measurements were calibrated using a gold sample with reference to the Au 4*f*~7/2~ and Au 4*f*~5/2~ lines located at 84 and 87.67 eV for the Al *Kα* radiation, respectively[39], according to the

Doniach-Sunjic method[40]. Throughout measurements, binding energy referencing and charge buildup corrections were made based on the prominent O 1$s$ core level from the substrate (located at 532.7 eV for quartz substrates and 532.6 eV for natively oxidized silicon substrates), rather than the commonly used C 1$s$ peak from adventitious carbon (adsorbed on the substrate), which can yield to misinterpretations[41,42]. The XPS data were analyzed using CasaXPS software. Unless otherwise specified, the HR XPS spectra were processed using Shirley backgrounds and fitted using Gaussian-Lorentzian line shapes. The C KLL Auger signal was processed using polynomial regression to generate the signal envelope, which was then differentiated to determine the D-parameter of patterned carbon microrods[43].

The morphology of the powder and the rods was evaluated using a scanning electron microscope (Hitachi TM 3000) set in COMPO mode for the backscattered electron detector, with e-beam in Analysis (15 kV) Standard Mode.

## Optical and electrical characterization of the carbon microstructures

PL spectral and PL lifetime decay measurements were carried out with a custom-built sample-scanning confocal microscope paired with a 466 nm pulsed diode laser (Picoquant LDH-P-C-470) at a repetition rate of 5 MHz. Signal acquisition involved the collection of PL spectra (PI Acton SP-2156 spectrograph coupled to EM-CCD, Andor iXon EM + DU-897 BI) and PL decays by time-correlated single photon counting (TCSPC)[44]. For TCSPC measurements, photons were collected with a fast single photon counting detector (Micro Photon Devices, PDM 50ct). This detector was also used for PL imaging. Photon timing was measured using a PDL 800-D pulsed laser driver that provided the timing signal to a PicoHarp 300 TCSPC module in combination with a PHR 800 detector router, all from Picoquant. All decay curves were tail-fitted with bi-exponential functions and yielded reported characteristic decay times $\tau_1$ and $\tau_2$[44]. To avoid sample degradation during testing, the samples were carefully sandwiched between two borosilicate glass cover slips under inert atmosphere.

## Single carbon microrod-based sensors

A commercial 120 Ω strain sensor (Omega,SGD-4/120-LY43, 5.7 mm × 3.8 mm) was attached to a polyester hinge. Long (>1 mm) carbon rods were attached using carbon paint. Individual rods were fastened to pads on a TSSOP-16 breakout board with electroless nickel immersion gold (ENIG) plated pads. The pads were 0.29 mm wide with a pitch of 0.7 mm on a flexible polyimide substrate. The assembly was mounted over the commercial strain gauge on the polyester hinge in a small vise enabling simultaneous response measurements from the commercial and microrod sensors.

Positive curvature was induced by moving the jaws of the vise closer together while pushing on the end of the polyester support. The change in resistance for each sensor was monitored in a quarter Wheatstone bridge configuration with a multiturn potentiometer as a balancing resistor. The excitation voltage was limited by the low resistance of the commercial fiber. An excitation voltage of 4 V DC was applied and the potentiometer was adjusted on each strain sensor until their respective bridge potentials were near 0. The potentials across each bridge were measured at 10 Hz while flexing the microrod/strain gauge assembly. Data was acquired using National Instrument Labview interfaced to a SCXI-1600 16-bit digitizer and a SCXI-1520 8-Channel Strain/Bridge Module equipped with a SCXI-1314 Universal Strain Terminal Block.

## Data availability

Source data are provided with this paper. Any additional data in the main text or the Supplementary Information is available upon request to the corresponding authors by explaining the files and format required. Source data are provided with this paper.

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

## Acknowledgements

We acknowledge the University of Central Florida's MCF-AMPAC and NanoScience Technology Center shared facilities, and the machine shop at the University of Central Florida. We acknowledge Saint-Gobain for providing h-BN. We acknowledge Prof. Rajaraman for providing the PDMS and insightful discussions about 3D printing. We acknowledge Dr. Lasri for the initial transfer and preparation of the structures. F.E.T.D., R.G.B., and L.T. acknowledge support from NSF CHE-1465105. F.E.T.D. and L.T. also acknowledge support from NSF CHE-1847830. L.T. acknowledges support from the Gordon and Betty Moore Foundation (10.37807/GBMF11568). The team acknowledges NSF MRI ECCS-1726636 and NSF MRI DMR-1920050.

## Author contributions

R.G.B., N.R. and L.T. conceptualized the study. F.E.T.D., R.G.B. and L.T. worked on the methodology related to carbon growth. F.E.T.D., E.E.B., K.L.C., S.S., M.M., L.R.S., L.T. and R.G.B. acquired the experimental data. All authors contributed to the investigation through continuous discussions. R.G.B., N.R. and L.T. contributed to the validation and formal analysis of the data. R.G.B. and L.T. supported and managed the work. T.J., A.G., R.G.B. and L.T. supervised the students involved in the work. R.G.B., N.R. and L.T. wrote the initial manuscript. All authors reviewed and edited the manuscript.

## Competing interests

The authors declare no competing interests.
