## [Peer Review File · Nature Communications]

Room temperature 3D carbon microprintingEditorial Note: Parts of this Peer Review File have been redacted as indicated to remove third-party material where no permission to publish could be obtained.

REVIEWER COMMENTS

Reviewer #1 (Remarks to the Author):

In this work done by Fernand et al., a low-power visible light-assisted approach was demonstrated for the growth of carbon microstructures at room temperature. 3D carbon micropatterning of well-controlled arrays or complex assemblies could be produced. The luminescence properties and ohmic behavior of the carbon materials were evaluated. The progress made herein bring benefits in manufacturing custom 3D functional materials. However, unsolved issues as shown below made the work less attractive and reject was recommended.

1) In the introduction part, advantages and disadvantages of the current approaches towards 3D carbon materials production was not well discussed and compared with the approach developed herein, as well as the properties of the carbon products.

2) Low carbon footprint was demonstrated as one of the merits in the developed approach. The production of BN catalysts was not considered.

3) The carbon materials obtained herein was in fact heterostructures composed of the BN catalysts. How to separate the performance of carbon and BN in diverse applications?

4) Although metal-free BN catalyst could be a strength in this work compared with the metal-promoted procedure, metal catalysts could be easily removed by acid-assisted washing, while BN is difficult to be moved.

5) How about the porosity the as-afforded carbon materials?

6) How about the optical and electrical properties of the carbon materials in this work compared with that of the carbons being obtained in the reported procedure?

Reviewer #2 (Remarks to the Author):

This paper reported a novel approach for the controlled growth of carbon microstructures at room temperature using low-power visible light. This process is enabled by a metal-free catalyst, defect-engineered hexagonal boron nitride, and yields carbon structures with illumination-dependent features. The authors demonstrate the ability to create complex 3D structures directly by orienting the incident light during growth. The patterned structures exhibit appealing luminescence properties and ohmic behavior, with great potential for optoelectronics and sensing applications, including those interfacing with biological systems. While the authors propose a compelling concept, they have not furnished proof

of three-dimensional printing of arbitrary shapes, a central tenet of 3D printing. Additionally, the lack of demonstrated real-world applications suggests that the manuscript might not be ready for publication in its current form. Besides, some other questions are listed below:

1. The paper insufficiently highlights the significance of carbon-based materials in 3D printing. Moreover, when it comes to electronic devices, the essential factors of mechanical stability and electronic properties of the materials are not comprehensively demonstrated.
2. The actual components of the hydrocarbon gas should be provided.
3. The authors ought to elucidate the adhesion properties of the carbon rod when applied to various substrates.
4. Despite the carbon formation occurring at room temperature, the reaction should ideally be conducted in a specially designed reactor. This requirement introduces an additional challenge due to the complexity associated with achieving a proper seal.
5. Considering the multi-branched microstructures attained through consecutive segmented growth from the active tip, what is the greatest turning angle that can be realized?
6. Many 3D printed parts require some level of post-processing, like support removal, sanding, or painting. This step is crucial for achieving a professional finish. How is the hBN removed without affecting the carbon rods?
7. The authors are encouraged to elaborate on the distinctive properties of their hBN catalyst, especially in comparison with traditional metal-based catalysts. Greater detail about the catalytic characteristics of their chosen material would be beneficial.
8. Could the authors comment on the consistency in the compositions of the carbon rod across different areas and different production batches?

Reviewer #3 (Remarks to the Author):

The authors report an approach for controlled, low-power visible light-assisted growth of carbon microstructures at room temperature on a wide range of substrates.

The results presented by the authors are very interesting and promising

The ability to create customized complex 3D structures directly by orienting the incident light during growth is praiseworthy.

However, certain information is missing.

(1) Which hydrocarbon has been used?

(2) Is the growth possible with any hydrocarbon?

(3) What is the role of pressure for adsorption of hydrocarbon onto Dh-BN

(4) Further clarification is needed to understand how precisely focusing the illumination up to 60 μm above the surface of the catalyst contributes to the growth of carbon micro rods in this process.

The answers to the above queries are required before publication

Reviewer #4 (Remarks to the Author):

Paper Review: Room temperature 3D carbon microprinting

Overall Rating: Major Revisions

Significance: High

Novelty: High

Broad Interest: Very high

Scholarly Presentation: High

Comments to the authors: The authors describe a new printing technique for the generation of high-aspect-ratio carbon microstructures at high resolution. The technique leverages defect-engineered boron nitride, which, after ball milling, has its band gap reduced to allow for population of the excited-state with visible light. The authors show that irradiation of this defect-engineered boron nitride (dh-BN) in a carbon-rich atmosphere leads to carbon growth along the axis of irradiation. The growth continues as irradiation continues until the laser light is focused too far out-of-plane. The authors demonstrate functionally continuous growth by moving the focal axis of the laser alongside the growth plane, resulting in extremely high-aspect-ratio carbon microstructures that are, to my knowledge, inaccessible through other printing or growth methods. The carbon ranges from graphitic (in the edges) to graphenic (in the core) based on Raman and PL spectra.

The strategy of using dh-BN for photocatalytic applications is not necessarily novel, but it has not achieved such striking results in printing previously, nor has it been applied for catalytic carbon deposition by dehydrogenation. Therefore, the novelty of the work is strong. Moreover, since the work presents a printing technique that allows for the generation of previously inaccessible structures, it has a large potential for impact. However, this substantial potential for impact depends on the performance of the synthesized high-aspect-ratio carbon being noticeably enhanced due to the printing methodology. Therefore, I would like to see the authors demonstrate an application for the printed materials, such as a supercapacitor or a sensor. Pending this and several other revisions described below, this manuscript can be considered for publication in Nature Communications.

Major Comments:

- The increased interlayer spacing found in XRD may also be due to turbostratic alignment of graphenic layers within the material matrix. See: Lin, J. et al., Nat Commun., 2014, 5, 5714.

o To test this hypothesis, authors should fit the 2D band of the material and see if it matches a single Lorentzian. Also present the full width at half maximum and look for the TS1, TS2, and M peaks in the Raman spectrum. See: Malard, L. M. et al., Phys. Rep., 2009, 473, 51-87. Also see: Garlow, J. A. et al., Sci. Rep., 2016, 6, 19804.

- Ideally, the authors should present a possible device, such as a supercapacitor or a sensor, leveraging the unique properties of their material. This would show the utility and quality of the synthesized material.
- What are the mechanical properties of these films? If h-BN coatings degrade, do these fibrous attachments also degrade? Is this stability a hindrance for potential applications?
- Much more experimental detail is needed to replicate the study. What is the identity of the growth gas? What is the operating pressure? How much catalyst is employed, and how is it adhered to the surface of substrates?
- To further accentuate the environmentally friendly and scalable nature of the process, a life cycle assessment would be useful.
- Does it matter how the hBN layer is deposited? (CVD, sputter coating, etc).

Minor Comments:

- Is it really appropriate to call this “room temperature”? A 100 mW laser focused on a single spot for several minutes will likely cause a temperature spikes. It might be more appropriate to call it photocatalytic.
- The electrical properties should be presented as the electrical properties of long carbon fibers, since these are the only constructs from which conductivity measurements could be extracted. The properties of the smaller fibers are likely to be different.
- “Details on data analysis are published elsewhere.” Pg 17 – where are these details published?
- Fig. 2a label core and shell – unclear at first which was which.
- How many contacts were used to measure resistance? Is there a chance contact resistance is affecting measurement?
- The main text claims a 2 mm rod, but Supplementary Fig. 16 shows the longest length rod in the manuscript is 1.5 mm.
- Does substrate material matter to the synthesis/stability of rods?
- The abstract should be revised so that the high aspect ratio of the generated structures is presented earlier and more prominently, seeing as how this is the major “novelty” and advance of the presented work.
- The authors should cite additional landmark references comparing their printing methodology to other methods in the literature for 3D carbon structure generation.
- Can you achieve carbon growth just by heating this chamber with the catalyst and gas, or does the excited state truly need to be populated? (This may be a difficult experiment to achieve, but I would like to see the authors’ thoughts on the question).

Response to referees' comments

We thank the referees for their thorough reviews of our manuscript. We provide below a point-by-point response to the comments. In addition, we have modified the manuscript according to the changes suggested or requested by the reviewers and the editor.

Referee #1 (Remarks to the Author):

In this work done by Fernand et al., a low-power visible light-assisted approach was demonstrated for the growth of carbon microstructures at room temperature. 3D carbon micropatterning of well-controlled arrays or complex assemblies could be produced. The luminescence properties and ohmic behavior of the carbon materials were evaluated. The progress made herein bring benefits in manufacturing custom 3D functional materials. However, unsolved issues as shown below made the work less attractive and reject was recommended.

1) In the introduction part, advantages and disadvantages of the current approaches towards 3D carbon materials production was not well discussed and compared with the approach developed herein, as well as the properties of the carbon products.

Response: We have revised the introduction of the manuscript to clarify this point, while trying to maintain a word count that is acceptable for the journal.

2) Low carbon footprint was demonstrated as one of the merits in the developed approach. The production of BN catalysts was not considered.

Response: The approach we present in this study only requires a ~10 mW illumination with a CW 532 nm laser, operated at room temperature for the carbon structures to grow. From the information available in the literature, producing the *h*-BN catalyst is expected to require significantly less energy than mining and processing noble metals such as platinum (Pt), palladium (Pd) or transition metals such as nickel (Ni) (Table R1). For large-scale production, the literature reports that one common synthesis route of *h*-BN uses melamine (an inexpensive polymer), water, urea, and boric acid. The process usually requires a few hours of heat treatment above 1000 °C, which adds to the energy cost of the process. Defects are introduced by ball milling for 2 hours, which is also an energy cost to take into account but constitutes a lower power draw than other defect engineering techniques such as heat treatments. The power consumed as a function of mass of material produced is addressed in by Blair, R. in Mechanical and Combined Chemical and Mechanical Treatment of Biomass, in Production of Biofuels and Chemicals with Ultrasound, 269-288, 2015. However, *h*-BN is being considered by several large manufacturers, including Saint-Gobain (<https://www.bn.saint-gobain.com/sustainability>), as an essential material to help reach carbon neutrality by 2050, suggesting that it is relatively sustainable in the current context. In addition, the cost of *h*-BN is significantly lower than noble metals, and the availability of borates makes it a more sustainable solution in terms of the geopolitical context.

Table. R1 | Raw materials involved in the production of catalysts, corresponding costs, manufacturing CO₂ emissions, and sources.

Material	CO ₂ emission	Cost per kg	Source
Pt	20,600 tons of CO ₂ emitted per one ton of Pt mined and refined	~\$29,000	South Africa, Russia
Pd	3880 tons of CO ₂ emitted per one ton produced	~\$36,500	Russia, Canada, South Africa, USA
Ni	18 tons of CO ₂ emitted per one ton produced 11.53 kg CO ₂ eq. per kg	~\$20	Russia, Indonesia, Philippines, Colombia, Canada
Borates - Boric acid - Borax	0.72 kg CO ₂ eq. per kg 1.65 kg CO ₂ eq. per kg	~\$10	Turkey, USA
h -BN		~\$30	

3) The carbon materials obtained herein was in fact heterostructures composed of the BN catalysts. How to separate the performance of carbon and BN in diverse applications?

Response: According to our XPS, Raman, and FTIR data, the structures are mostly composed of carbon, with residues (*i.e.*, nanoscale clusters) of the BN catalyst. We provide additional information on the properties of the structures throughout the response.

Hereafter, we show that the properties of the as-synthesized structures are suitable for several applications including microelectrodes, strain sensors, and temperature sensors. Conventional carbon fibers have also been used for strain sensors (Wang et al., Journal of Materials Research, 14, 790–802, 1999; Mäder et al., Proceeding of the 18th International Conference on Composite Materials, Jeju Island, Korea, 21–26, 2011) and temperature sensors (Forintos et al., Composites Part A: Applied Science and Manufacturing, 131, 105819, 2020).

In a sense, the presence of BN residues in the shell and tip expands the realm of applications of carbon-based fibers to the fields of catalysis and optoelectronics, without the need for further treatment/doping of the carbon material, though these have not been comprehensively tested yet. However, these studies will require measurements that are beyond the scope of this study.

4) Although metal-free BN catalyst could be a strength in this work compared with the metal-promoted procedure, metal catalysts could be easily removed by acid-assisted washing, while BN is difficult to be moved.

Response: We have considered this aspect. Our observations and measurements to date suggest that BN is mostly contained in the tip of the microrods. XPS analyses (Supplementary Fig. 14 and corresponding text) show that B and N are not readily detected when characterizing

the shell of the microrod. Nanoscale imaging of the shell suggests that isolated clusters of BN are present and sparsely distributed in the carbon-rich matrix.

Hence, removing the tip of the rod is expected to remove most of the catalyst from the structures. We demonstrate this with the laser technology available to us at the time of this response. We show in Fig. R1 that the rod can be sectioned with a 532 nm laser at laser power of 10 mW focused with a 10× objective for a few seconds. The scanning electron microscopy (SEM) image in Fig. R1b shows the morphology of the sectioned region of the rod. More advanced techniques, such as those used to section carbon nanotubes, can be implemented for a cleaner result. For applications requiring better control of the section, focused ion beam (FIB) can be used. An example of a section obtained by FIB is provided in the inset of Fig. 2a. Others are shown in Fig. R3. It has been observed that boron nitride is impervious to acids but etched by strong bases. Both concentrated aqueous (Lee et al. Nano Letters, 15(2), 1238-1244, 2015) and molten hydroxide eutectics (Edgar et al. Journal of Applied Physics, 122(22), 225110, 2017) etch *h*-BN. Soaking the rod in 6 M sodium hydroxide for 24 hours did not affect the morphology of the structures.

Fig. R1 | Images of the carbon microrod sectioning. **a**, Optical image of a carbon structure section using a focused 532 nm laser. The dotted red line indicates the motion of the laser. Frames (i), (ii) and (iii) demonstrate successive sectioning of the rod. Scale bars represent 100 μm . **b**, SEM image of the sectioned structure shown in **a**. Scale bar represents 1 μm .

We note that BN may not have to be removed depending on the application. For instance, we show in Figs. R6, R11 and R12 that the rods can be used as a microelectrode to acquire I-V curves, as a strain sensor and as a temperature sensor, without the need to remove *h*-BN residues from them.

5) How about the porosity the as-afforded carbon materials?

Response: We carried out nanoscale imaging of the carbon structures. We imaged both the surface with atomic force microscopy (AFM) (Fig. R2) as well as the cross sections and longitudinal sections of the rods obtained by FIB (Fig. R3). In both cases, the material appears to be dense compared to structures referred to as porous carbon fibers in the literature (Gan et al., ChemEngineering, 4(4), 59, 2020).

Fig. R2 | AFM topography image (8 μm × 16 μm) of a carbon structure with corresponding high-resolution AFM topography image of the shell of the structure.

Fig. R3 | High-resolution image of FIB sectioned 3D printed carbon structures. a–d, Images of the texture of the core of the cross sectioned (a,b) and longitudinally sectioned (c,d) carbon microstructures.

6) How about the optical and electrical properties of the carbon materials in this work compared with that of the carbons being obtained in the reported procedure?

Response: We previously reported the optical and electrical properties of the carbon structures in Fig. 4 of the manuscript. The structures were found to be conductive in the case of the long rods characterized, as shown in Fig. 4d. The rods isolated for optical characterization exhibit some photoluminescence properties presented in Figs. 4a–c.

We carried out control measurements on commercial carbon fibers, graphite and on the standard carbon paint used to connect the structures to the gold (Au) pads for our electrical measurements. The materials were mounted across hot air solder leveling (HASL) coated test pads spaced 0.7 mm on a FR-4 substrate. The test pads were connected via pin headers to an HP 6114A precision power supply and a 1000 Ω precision resistor in a voltage divider configuration. The voltage drop across the 1000 Ω resistor was used to determine the current at any given voltage. The voltage drop was determined by two-point probe measurements with a FLUKE 117 True RMS multimeter.

The I-V and power curves are presented in Fig. R4. The measurements suggest that the electrical resistance of the long 3D printed carbon microrod is about 10 times higher than that of the commercial carbon fiber tested. However, we note that our carbon structures are analyzed as synthesized, and it is expected that the resistance could be reduced by further treatment of the rods after production, as is done when pyrolyzing carbon fibers. These measurements have been added to the Supplementary Information (SI) (see Supplementary Fig. 18).

Fig. R4 | I-V curves (top) and corresponding power curves (bottom) of graphite (red), carbon paint used for contacts (blue), commercial carbon paint (yellow), and long 3D printed carbon rods (black).

Referee #2 (Remarks to the Author):

This paper reported a novel approach for the controlled growth of carbon microstructures at room temperature using low-power visible light. This process is enabled by a metal-free catalyst, defect-engineered hexagonal boron nitride, and yields carbon structures with illumination-dependent features. The authors demonstrate the ability to create complex 3D structures directly by orienting the incident light during growth. The patterned structures exhibit appealing luminescence properties and ohmic behavior, with great potential for optoelectronics and sensing applications, including those interfacing with biological systems. While the authors propose a compelling concept, they have not furnished proof of three-dimensional printing of arbitrary shapes, a central tenet of 3D printing. Additionally, the lack of demonstrated real-world applications suggests that the manuscript might not be ready for publication in its current form. Besides, some other questions are listed below:

1. The paper insufficiently highlights the significance of carbon-based materials in 3D printing. Moreover, when it comes to electronic devices, the essential factors of mechanical stability and electronic properties of the materials are not comprehensively demonstrated.

Response: We have revised the introduction and conclusion of the manuscript to highlight the significance of 3D printing of carbon-based materials.

We provide below additional information regarding the mechanical stability and electronic properties of the printed carbon materials.

We show in Fig. R5 that the carbon structure can undergo significant bending without damage. The radius of the structures in Fig. R5a is $\sim 230 \mu\text{m}$, which is similar to the bending radii reported for carbon fibers in the literature before rupture (Loidl et al., Phys. Rev. Lett., 95, 225501, 2005). The configuration presented in Fig. R5 corresponds to a strain of $\sim 1.06\%$, which is well above reported values for carbon fibers (Naito et al. ICCM-17 - 17th International Conference on Composite Materials, Edinburgh, UK, 2009).

Here, the elasticity of the rod allows full recovery of its original shape upon releasing the constraint, as shown in Movie R1 provided as a separate file for this response. SEM images of the same carbon microrod while bent (Fig. R5b) and after releasing the constraint (Figs. R5c,d) show that the deformation does not affect the morphology.

Fig. R5 | Elasticity and mechanical stability of the carbon microrods. **a**, SEM image of a 3D printed carbon microrod bent into a loop. The inset shows the optical view of the structure. **b**, Higher resolution SEM image of the region of the carbon fiber marked in **a**. **c**, SEM image of the same structure after releasing the loop. **d**, Higher resolution SEM image of the rod in **c** showing no morphological damage due to bending. The movie corresponding to this process is available in Movie R1.

Next, we show that the structures can be used as an electrode for electrical measurements without damaging the electrode during its deformation due to steps of manipulation and displacement over the different regions of the substrates. Individual fibers were mounted on a silicon chip to facilitate precise manipulation. The fiber was grounded while the voltage ramp was applied to the substrate in this setup. The tip of the fiber was approached to the surface of different materials, including Au, silicon, PVA, and biological cells. I-V curves were collected with the carbon fiber as an electrode. The results are presented in Fig. R6.

Additional examples of the behavior of the structures are presented hereafter. The devices developed include a strain sensor (Fig. R11) and a temperature sensor (Fig. R12) using a single carbon microrod.

Fig. R6 | Carbon microrods used as electrode-probes for electrical measurements. **a,b**, Optical views of the carbon microelectrode held on an AFM cantilever holder away from the surface (**a**) and in contact with the sample surface (**b**). **c–f**, I-V curves collected on different materials with the microelectrode away from the surface (**c**), on Au (**d**), on Si (**e**), and on a fixed mammalian cell nucleus and cytoplasm (**f**).

2. The actual components of the hydrocarbon gas should be provided.

Response: For all the data presented in this manuscript, propene was used as the hydrocarbon gas. This has been clarified in the Methods of the revised manuscript.

3. The authors ought to elucidate the adhesion properties of the carbon rod when applied to various substrates.

Response: Qualitatively, the structures are anchored on the catalyst through their so-called “roots”, as shown in Fig. 1g and Supplementary Fig. 4. The mechanical removal of the carbon structures usually includes the “roots”, which can later be removed by gentle pressure using precision microtools. In the case of thinner coatings, such as on the Kevlar fibers, the carbon structures were more strongly anchored. In fact, the multiple rods grown in the Kevlar fiber shown in Fig. 1e remained intact for several months despite various manipulations of the sample and storage in air.

We evaluated the adhesion properties of the carbon rods by attaching individual rods to an AFM cantilever followed by acquisition of force curves on various substrates. The results are presented below (Fig. R7) and show that the nature of the interactions between the carbon structure and the substrate vary. In the case of PDMS, a long-range attractive force was evident from the approach curve, which may be attributed to the –OH groups at the surface of the sample. No long-range interaction was measured during the interaction with Au, Si, and SiO₂.

The force curve collected when withdrawing the structure away from the substrate indicates a very large adhesion between the rod and Au compared to Si and SiO₂.

Fig. R7 | Nature of the interaction between the carbon microrods and various substrates. Approach (red) and withdraw (blue) curves representing, qualitatively, the nature of the interaction between the shell of the carbon structure and SiO₂, Si, Au, and PDMS. The strongest adhesion is observed with Au, while the interaction with PDMS suggests a long-range attractive force with the carbon rod.

4. Despite the carbon formation occurring at room temperature, the reaction should ideally be conducted in a specially designed reactor. This requirement introduces an additional challenge due to the complexity associated with achieving a proper seal.

Response: The reaction is indeed conducted in a reactor, as shown in Supplementary Fig. 2. However, the design of the reactor is rather simple, with a sapphire window to give laser access to the surface, and an input and output for gas injection and purging, respectively. The design presented in Supplementary Fig. 2 was machined in-house, and the parts required for the window and connectors provided a low-cost prototype, under 100 \$. We note that this could be further engineered for large scale implementation, using a small gas injector upon printing, which would bypass the need for the reactor. We found that carbon growth can also be achieved at close to atmospheric pressure if the catalyst is pressurized/saturated overnight with the gas prior to illumination.

5. Considering the multi-branched microstructures attained through consecutive segmented growth from the active tip, what is the greatest turning angle that can be realized?

Response: The angle demonstrated in the structure presented in the manuscript is $\sim 90^\circ$. Due to limitations of the motors and space available on our current platform, we could only demonstrate an angle of $\sim 3^\circ$, $\sim 8^\circ$ and $\sim 90^\circ$ (Fig. R8). However, we expect that freeform designs will become possible with a 5-axes printing system.

Fig. R8 | Multibranched 3D structures. **a**, SEM image of carbon microstructures grown with a 3° angle with respect to the substrate surface normal. **b**, SEM image of branched carbon microstructures grown with secondary structures grown with 3° and 8° angles with respect to the initial vertical carbon segment. **c**, Optical image of the branched structures with secondary branches grown at a 90° angle with respect to the initial segment.

6. Many 3D printed parts require some level of post-processing, like support removal, sanding, or painting. This step is crucial for achieving a professional finish. How is the hBN removed without affecting the carbon rods?

Response: In this work, the quantities of *h*-BN contained in the carbon rods are very small, sometimes undetectable. Some applications may suffice with the rods as-synthesized without the need for removal, as we showed that the structures are conductive and present high mechanical performances. The high aspect ratio of the structures can be appealing for sensing applications, or measurements in liquid environments. The surface of the rod is very smooth as synthesized (see Figs. R1, R2, and R5). We verified that the rods could be coated with a thin gold layer. Such coating could be used if the presence of *h*-BN residues turns out to be an issue, or with a thin film of insulating material to isolate the fiber and expose only the conducting tip for measurements in liquid for instance. The structure could be further processed as needed for the applications considered. If some applications require *h*-BN removal, laser or FIB sectioning can be considered, as shown in Figs. R1 and R3.

7. The authors are encouraged to elaborate on the distinctive properties of their hBN catalyst, especially in comparison with traditional metal-based catalysts. Greater detail about the catalytic characteristics of their chosen material would be beneficial.

Response: The catalytic properties of *h*-BN are engineered by introducing defects in the *h*-BN lattice. In this work, the defects were introduced by ball milling. XPS analysis of the milled powder suggests that the treatment predominantly introduces nitrogen vacancies in the lattice. Recent work by our team suggests that other processes of introducing defects in the lattice are also viable to achieve 3D-printing of carbon structures, which will be presented elsewhere.

8. Could the authors comment on the consistency in the compositions of the carbon rod across different areas and different production batches?

Response: The consistency of the length, diameter, surface roughness, fluorescence emission and Raman fingerprint of the structures suggest that the process offers high reproducibility within the same batch and from batch to batch, given that experimental parameters are identical. These parameters include thickness of the catalyst, laser illumination power and wavelength, magnification, position of the focal point with respect to the surface, gas pressure, temperature, and speed of motor displacement.

Referee #3 (Remarks to the Author):

The authors report an approach for controlled, low-power visible light-assisted growth of carbon microstructures at room temperature on a wide range of substrates.

The results presented by the authors are very interesting and promising

The ability to create customized complex 3D structures directly by orienting the incident light during growth is praiseworthy.

However, certain information is missing.

(1) Which hydrocarbon has been used?

Response: For all the data presented in this manuscript, propene was used as the hydrocarbon gas. Methods of the manuscript have been updated.

(2) Is the growth possible with any hydrocarbon?

Response: The growth is possible with other hydrocarbons (methane, propene, allene) as well as other carbon-based gases such as CO, by adapting experimental conditions.

(3) What is the role of pressure for adsorption of hydrocarbon onto Dh-BN

Response: Our measurements suggest that lowering the pressure affects the shape, length and width of the rods, as shown in Fig. R9. Although we could not carry out enough measurements to develop a model in the time allotted for the revisions, we infer that mass transport and surface-reaction kinetics should be considered during growth, and conditions to favor one or the other should be determined for different experimental conditions. Observation of luminescence around the structure during growth suggest that gas-phase nucleation could also play a role in the process.

Fig. R9 | SEM images of carbon microstructures obtained at propene pressures varying from 100 to 10 psi (~690 to 69 kPa). Growth was obtained with a 10× objective with 532 nm illumination at 100 mW. Scale bars represent 5 μ m.

(4) Further clarification is needed to understand how precisely focusing the illumination up to 60 μm above the surface of the catalyst contributes to the growth of carbon micro rods in this process.

Response: Our observations suggest that growth likely involves a multi-photon process, which can only be satisfied by focusing the light on the experimental system we used for the present work. A 10 \times objective with numerical aperture NA of 0.3 exhibits a focal volume with $r_{xy} \sim 1 \mu\text{m}$ and $r_z \sim 11 \mu\text{m}$ based on approximations using Abbe's formula. As can be seen in the measurements presented in this work, the dimensions of the rods exceed the size of the focal volume, which suggests that regions with lower photon density satisfy the conditions for growth to an extent. Laterally, the structures vary from $\sim 5\text{--}8 \mu\text{m}$ in diameter, which suggests a region of illumination satisfying the conditions for growth about 5–8 times larger than the radial dimension of the focal volume r_{xy} . Assuming that the same is true in the z-direction, the region with suitable photon counts reaching the defect site at instant t is satisfied as long as the focal distance is within $\sim 50\text{--}80 \mu\text{m}$ above the surface. Our assumption is that scattering hinders the multi-photon process when the focal volume is within the catalyst layer.

The answers to the above queries are required before publication

Referee #4 (Remarks to the Author):

Paper Review: Room temperature 3D carbon microprinting

Overall Rating: Major Revisions

Significance: High

Novelty: High

Broad Interest: Very high

Scholarly Presentation: High

Comments to the authors: The authors describe a new printing technique for the generation of high-aspect-ratio carbon microstructures at high resolution. The technique leverages defect-engineered boron nitride, which, after ball milling, has its band gap reduced to allow for population of the excited-state with visible light. The authors show that irradiation of this defect-engineered boron nitride (*dh*-BN) in a carbon-rich atmosphere leads to carbon growth along the axis of irradiation. The growth continues as irradiation continues until the laser light is focused too far out-of-plane. The authors demonstrate functionally continuous growth by moving the focal axis of the laser alongside the growth plane, resulting in extremely high-aspect-ratio carbon microstructures that are, to my knowledge, inaccessible through other printing or growth methods. The carbon ranges from graphitic (in the edges) to graphenic (in the core) based on Raman and PL spectra.

The strategy of using *dh*-BN for photocatalytic applications is not necessarily novel, but it has not achieved such striking results in printing previously, nor has it been applied for catalytic carbon deposition by dehydrogenation. Therefore, the novelty of the work is strong. Moreover, since the work presents a printing technique that allows for the generation of previously inaccessible structures, it has a large potential for impact. However, this substantial potential for impact depends on the performance of the synthesized high-aspect-ratio carbon being noticeably enhanced due to the printing methodology. Therefore, I would like to see the authors demonstrate an application for the printed materials, such as a supercapacitor or a sensor. Pending this and several other revisions described below, this manuscript can be considered for publication in Nature Communications.

Response: We thank the reviewer for the impactful summary of the work. We note that even after a recent literature survey, we cannot find any publication demonstrating the photocatalytic applications of *h*-BN with visible light. All studies presented in a recent review on the photocatalytic activity of boron nitride-based materials indicates that UV light (< 320 nm) is required to observe a reaction of *h*-BN alone, or adding another compound is required to shift the reaction toward the visible range (Laghaei et al. J. Mater. Chem. A, 11, 11925–11963, 2023).

Major Comments:

- The increased interlayer spacing found in XRD may also be due to turbostratic alignment of graphenic layers within the material matrix. See: Lin, J. et al., Nat Commun., 2014, 5, 5714.

To test this hypothesis, authors should fit the 2D band of the material and see if it matches a single Lorentzian. Also present the full width at half maximum and look for the TS1, TS2, and M peaks in the Raman spectrum. See: Malard, L. M. et al., Phys. Rep., 2009, 473, 51-87. Also see: Garlow, J. A. et al., Sci. Rep., 2016, 6, 19804.

Response: We investigated the Raman 2D (or G') band further based on this interesting suggestion. As shown in Fig. R10, a single Lorentzian function provided a good fit of the 2D (or G') band in the Raman spectra acquired in the rod. The full width at half max of the peak was found to be $\sim 105 \text{ cm}^{-1}$. This suggests a turbostratic alignment of the graphenic layers in the structure. The TS1, TS2 and M peaks were not observed in our spectra.

Fig. R10 | 2D band of the Raman spectrum of the carbon structure with Lorentzian fit.

The manuscript has been modified accordingly, and the fit has been added to the SI.

- Ideally, the authors should present a possible device, such as a supercapacitor or a sensor, leveraging the unique properties of their material. This would show the utility and quality of the synthesized material.

Response: We have carried out additional measurements to demonstrate the suitability of the carbon structures as a strain sensor (Fig. R11) and a temperature sensor (Fig. R12), in addition to its suitability as an electrode (Fig. R6), which could be extended for electrocatalysis. The data has been added to the SI.

Application as a strain sensor: A single carbon microrod was adhered across electroless nickel immersion gold (ENIG) coated test pads spaced of 0.7 mm on a flexible polyimide substrate. The mounted assembly was affixed to a polyester hinge in a small vise. Positive curvature was induced by moving the jaws of the vise closer together while pushing on the end of the polyester support. The change in resistance was monitored in a quarter Wheatstone bridge configuration with a multiturn potentiometer as a balancing resistor. An excitation voltage of 5 V DC was applied and the potentiometer was adjusted until the bridge potential was near 0. The potential

across the bridge was measured in timed intervals while flexing the carbon microrod. Data was acquired using National Instrument Labview interfaced to a SCXI-1600 16-bit digitizer and a SCXI-1520 8-Channel Strain/Bridge Module equipped with a SCXI-1314 Universal Strain Terminal Block (Fig. R11).

Fig. R11 | a, Wheatstone bridge on a flexible polyimide substrate. **b**, Photograph of the flexed system used to measure the effect of strain on the resistance of the carbon microrod. **c,d**, Photographs of the carbon microrod on the device before (**c**) and during flexing (**d**). **e**, Effect of flexing the device on the bridge potential of the device. **f**, Effect of successive flexing cycles on the fractional change in the device electrical resistance.

In this study, we subjected the carbon microrod to a strain of 0.7 % leading to a gauge factor of -34.2. This is notable as it is superior to gauge factors of 2-4 commonly reported for strain gauges (Scholle et al., J. Compos. Sci. 5, 96, 2021). In addition, the strain achieved without damaging the structure is greater than the failure strain of some carbon fibers.

Application as a temperature sensor: A single carbon microrod was adhered across Hot air solder leveling (HASL) coated test pads spaced of 0.7 mm on a FR-4 substrate. The mounted assembly was placed in a custom optical cell with 4 electrical feedthroughs. Two were used to sense the fiber resistance and two were used for a PT100 RTD positioned underneath the test board. Temperature was controlled via optical heating through a sapphire window. The change in resistance was monitored in a quarter Wheatstone bridge configuration with a multiturn potentiometer as a balancing resistor (Fig. R12). An voltage of 5 V DC was applied and the potentiometer was adjusted until the bridge potential was near 0. The potential across the bridge was measured in timed intervals heating the fiber. Data was acquired using National Instrument Labview interfaced to a SCXI-1600 16-bit digitizer and a SCXI-1520 8-Channel Strain/Bridge Module equipped with a SCXI-1314 Universal Strain Terminal Block. The temperature was monitored via a PT100 thermocouple positioned under the test board. The RTD signal was amplified by a MAX31865 Temperature Sensor Amplifier and polled via USB. The fiber showed a significant temperature response for a temperature change of 100 °C.

Fig. R12 | a, Picture of the Wheatstone bridge setup used for our temperature sensor. **b**, Fractional change in the device electrical resistance as a function of increasing temperature.

- What are the mechanical properties of these films? If h-BN coatings degrade, do these fibrous attachments also degrade? Is this stability a hindrance for potential applications?

Response: Our observations suggest that the carbon structures are stable for almost a year, even when stored in air. We verified that the structures retain their morphological and electrical properties.

We have not observed any degradation of the *h*-BN coatings expect their loss of reactivity when exposing them to air. However, arrays of carbon structures can remain on the *h*-BN coating for several months without being disturbed. Separating the carbon structures from the *h*-BN coating can be done immediately after growth, or much later. We do not anticipate it being a hindrance for applications.

- Much more experimental detail is needed to replicate the study. What is the identity of the growth gas? What is the operating pressure? How much catalyst is employed, and how is it adhered to the surface of substrates?

Response: We have updated the Methods of the paper to provide all experimental details of the growth process. The results presented here are obtained with propene pressurized at 40 psi (276 kPa), on loose *dh*-BN deposited on various substrates with thicknesses varying from few nm (Supplementary Fig. 5) to 2 mm (Figs. 1a,b). In Fig. R9, we show the effect of pressure on the structure growth after pressurizing the powder at 100 psi (690 kPa) for a few hours, which suggests that pressure can be used as a parameter to control the morphology of the structure in this process.

The catalyst can be loosely deposited on any surface. For instance, the catalyst was placed in a 2 mm × 2 mm × 2 mm well to facilitate handling in the example shown in Figs. 1a,b. In other cases presented in the manuscript (fabric, quartz, etc.), a thinner layer of BN was adhered on the substrate by mechanically depositing the powder.

- To further accentuate the environmentally friendly and scalable nature of the process, a life cycle assessment would be useful.

Response: From the information available in the literature, a common synthesis route for large-scale production of *h*-BN involves melamine, water, urea, and boric acid. All components are projected to be less energy intensive to produce and more sustainable than the mining of noble metals (see notes about this point in page 1 of the response). All are also commonly produced at large scale already, with efforts underway to reduce the carbon footprint of their production.

The formation of *h*-BN involves few hours of heat treatment above 1000 °C, while the introduction of defects is done by ball milling *h*-BN for 2 hours. The production of *h*-BN is already scaled to industrial production. For instance, the use of ball milling to treat 2D materials at an industrial scale has been successfully demonstrated for graphene oxide.

The measurements we have carried out so far suggest that the formation of carbon structures could be obtained with directed light such as the laser illumination presented here. With power in the ~10–100 mW range, printing of the structures can be done in a manner that resembles 3D printing. Other illuminations could be used as long as sufficient photon counts per unit area can be reached for the carbon capture and conversion to take place. In such a case, the structures may be larger or smaller depending on the type of optical setup considered. Designs for carbon material formation can also be considered for powder-like materials. In this case, non-directional illuminations, such as arc lamps or solar illuminators, could be used. Our team has confirmed the feasibility of this process, which will be the topic of future studies.

Fig. R13 | Overview of the context of the study presented in this work.

- Does it matter how the hBN layer is deposited? (CVD, sputter coating, etc).

Response: In this work, carbon growth was obtained on defect-laden *h*-BN flakes deposited as a loose thin powder film on the substrate. We have observed carbon nanostructure growth on exfoliated *h*-BN although the size of the features remained in the nanoscale range. The process has not been optimized yet for these conditions.

It is expected that the loose *h*-BN powder in the tip of the carbon microstructures is essential for growth to be sustained with the focused light moving at the same rate as the tip displacement due to growth.

Minor Comments:

- Is it really appropriate to call this “room temperature”? A 100 mW laser focused on a single spot for several minutes will likely cause a temperature spikes. It might be more appropriate to call it photocatalytic.

Response: From our survey of the literature, most processes occurring without supplying an external source of heat are referred to as room temperature processes. In our case, the reactor was not heated, and growth occurred at laser power as low as 10 mW. Under these conditions, we confirmed that water was not evaporated in the time period required for the growth and that polymers such as polystyrene were not affected when the laser was focused on the surface. However, we can confirm that the reaction changes the local temperature of the region since polystyrene softened locally after the carbon growth was initiated. Having said that, this process involving an exothermic reaction is different from an external input of thermal energy.

- The electrical properties should be presented as the electrical properties of long carbon fibers, since these are the only constructs from which conductivity measurements could be extracted. The properties of the smaller fibers are likely to be different.

Response: We have modified the text for accuracy regarding this point.

- “Details on data analysis are published elsewhere.” Pg 17 – where are these details published?

Response: We have added the appropriate reference in the text.

- Fig. 2a label core and shell – unclear at first which was which.

Response: We have labelled the curves presented in Fig. 2a to clarify this point.

- How many contacts were used to measure resistance? Is there a chance contact resistance is affecting measurement?

Response: Two contacts were used to measure the resistance, as indicated in the Methods section. We used carbon paint for electrical resistance measurements to limit the contact resistance. As can be seen in Fig. R4, the resistance of the carbon paint and commercial carbon fibers measured with the same system suggest that the contact resistance is not playing a significant role in the resistance measured. The resistance of the carbon microrods in the 2-5 k Ω range is much larger than the contacts ($\sim 25 \Omega$) therefore making the two-contact resistance measurement sufficient.

- The main text claims a 2 mm rod, but Supplementary Fig. 16 shows the longest length rod in the manuscript is 1.5 mm.

Response: While carrying out new measurements for this revision, rods of up to 2.99 mm were produced, as shown in the optical image below (Fig. R14). We have added this figure to the Supplementary Information (Supplementary Fig. 17). The width of the longest rod was $\sim 6 \mu\text{m}$. Based on these new values, we also updated the aspect ratio that can be reach to ~ 500 in the text.

Fig. R14 | Optical image of 2.9 and 2.4 mm-long carbon rods obtained by continuously moving the focal point of the laser to drive growth with an illumination of 532 nm using a 10× objective and laser power of 10 mW. The speed of motion was set at 5 $\mu\text{m/s}$ for these rods.

- Does substrate material matter to the synthesis/stability of rods?

Response: We believe that the growth process on the catalyst is independent of the substrate if the layer of catalyst is thick. To date, we have demonstrated growth on catalyst only (Figs. 1a,b), and on thin films of catalysts on polymers (PDMS and polystyrene, Fig. 1f), textile fibers (Kevlar, Figs. 1c–e), Si (not shown), SiO₂ (Fig. 3), lithium niobate and quartz (rods produced for the XPS measurements and revisions).

- The abstract should be revised so that the high aspect ratio of the generated structures is presented earlier and more prominently, seeing as how this is the major “novelty” and advance of the presented work.

Response: We have revised the abstract to reflect this comment and the new aspect ratio achieved during growth.

- The authors should cite additional landmark references comparing their printing methodology to other methods in the literature for 3D carbon structure generation.

Response: The following references have been added:

Rife et al., Carbon, 162, 95–105, 2020.

Yuk et al., Nat. Commun. 11, 1604, 2020.

Fu, et al., Adv. Mater. 29, 1603486, 2017.

- Can you achieve carbon growth just by heating this chamber with the catalyst and gas, or does the excited state truly need to be populated? (This may be a difficult experiment to achieve, but I would like to see the authors' thoughts on the question).

Response: As discussed in page 4 of the SI, we have carried some experiments in a reactor used for thermal catalysis and could not observe any reaction when the same catalyst was pressurized under propene and heated to 500 °C. Our measurements suggest that populating the excited state is necessary since the growth process only occurs as photon energy is above 2 eV.

REVIEWER COMMENTS

Reviewer #1 (Remarks to the Author):

The authors addressed my concerns and accepted was recommended in its current status.

Reviewer #2 (Remarks to the Author):

The revised manuscript has undergone considerable amendments; yet, it remains apparent that the capability of 3D printing in arbitrary shapes has not been conclusively demonstrated. This is particularly evident from Fig. R8b, where alterations in the laser angle appear to compromise the uniformity of the fibers. Such observations suggest that the technique may be constrained to the fabrication of linear carbon fibers, limiting its applicability in more complex geometries. Conversely, the paper does not sufficiently demonstrate the adaptability of printing carbon fibers onto various substrates. The current description suggests that the fibers must be printed and then manually relocated for use in applications such as sensors. This two-step process raises questions about the practicality and efficiency of the technique in a real-world setting.

Additionally, the lack of a defined target application for the printing technology is notable. While the authors discuss the technique in a broad sense, a specific application context would provide a more concrete framework for comparing this new technique to existing ones. By identifying and targeting a particular application, the authors could offer clearer insights into the potential advantages or limitations of their approach when contrasted with established methods, thereby enhancing the relevance and impact of their research within the field.

Reviewer #3 (Remarks to the Author):

In the revised version of the paper the authors have clarified all the queries raised by me. I am convinced that the present version is suitable for publication in Nature Communications

Reviewer #4 (Remarks to the Author):

The authors have provided substantial data towards answering my concerns about the paper and explored the potential applications of the unique carbon microstructures their printing technique is capable of generating. However, a few more changes need to be made before this work rises to the

standards of significance commonly met by peer manuscripts published in Nature Communications.

First, my initial review asked for “a possible device... that leveraged the unique aspects of their system.” While, indeed, the high aspect ratio of the generated structures may lead to superior performance for the structures generated in strain sensors, etc., I would like to see the authors directly test this hypothesis by comparing against a common sensor material or other standard that does not benefit from the unique advantage of high aspect ratio conferred by the novel printing method presented by the authors. Most importantly, I would like to see data directly testing the hypothesis that the advantages conferred by the novel printing technique (i.e. high aspect ratio) are useful.

Second, the performance of the fabricated carbon microstructures should strongly support the significance of the work and therefore should be included as part of the main text. Without this aspect of the story featured prominently, again, I feel that the demonstrated impact of this new printing strategy could be viewed as a less significant advance.

REVIEWER COMMENTS

Reviewer #1 (Remarks to the Author):

The authors addressed my concerns and accepted was recommended in its current status.

We thank reviewer 1 for their review of the work.

Reviewer #2 (Remarks to the Author):

The revised manuscript has undergone considerable amendments; yet, it remains apparent that the capability of 3D printing in arbitrary shapes has not been conclusively demonstrated. This is particularly evident from Fig. R8b, where alterations in the laser angle appear to compromise the uniformity of the fibers. Such observations suggest that the technique may be constrained to the fabrication of linear carbon fibers, limiting its applicability in more complex geometries.

We provide here additional information to address this comment. The technique is indeed suitable to form complex geometries as mentioned in the manuscript and the previous responses. We have demonstrated in the manuscript the formation of 3D multi-branched structures with 90° angles between the branches. This is a significant accomplishment for 3D microscale printing in itself and has never been demonstrated for carbon structures at the scales described here, to the best of our knowledge.

To further address the added comment above, we carried out additional measurements that demonstrate the ability of changing the direction of growth while maintaining the width of the microstructures. The results are presented in Figure R1. The four examples of nonlinear microstructures demonstrate that the direction can be varied throughout the growth of the millimeter-long structure (Figure R1a,b), that the direction can be varied with different angles with the normal to the sample (Figure R1c), and that the direction change does not result in segmented features if the growth parameters are optimized appropriately (Figure R1d). We note that the microstructures presented in Figure R1 were transferred to a sapphire substrate to obtain the images to obtain more stable imaging conditions. Our work to implement a 5-axes printing system to control freeform printing, which is beyond the scope of this manuscript, will enable the real-time management of the laser direction and focal position in concert with the growth rate to produce custom design layout.

Fig. R1 | Images of non-linear carbon microrods. a–d, Scanning electron microscopy images of four different carbon microstructures grown on glass with varying directions of growth.

Conversely, the paper does not sufficiently demonstrate the adaptability of printing carbon fibers onto various substrates. The current description suggests that the fibers must be printed and then manually relocated for use in applications such as sensors. This two-step process raises questions about the practicality and efficiency of the technique in a real-world setting.

As previously indicated, we have achieved printing the carbon microstructures on several substrates in the manuscript including glass (Figure 3a,b,d,e), quartz (Figure 2), silicon, Kevlar (Figure 1c,d,e), PDMS (Figure 1f). In addition, we have prepared new samples to clearly illustrate that the fibers can be printed on the substrate coated with the catalyst, and manipulated as-is. We present some examples of microstructures grown on Kapton tape (Figure R2a–c), gold electrodes (Figure R2a–c) and Kevlar fibers (Figure R2d–f).

Fig. R2 | Images of carbon microrods printed on various substrates. a–c, Scanning electron microscopy images of carbon microstructures grown on Kapton tape (dark region) and gold (bright region) electrodes. **d–f,** Scanning electron microscopy images of carbon microstructures grown on Kevlar fibers.

All the sample prepared can be transported and manipulated as-is with rods remaining attached to the substrate, as was the case for the samples imaged in Figure R2 (substrate was transported across campus

with no particular precaution in this case). However, as can be seen in Figure R2e, imaging the carbon microstructures on the substrate coated with h-BN can result in unwanted contamination of the structures in presence of the electron beam, which can hide important features of interest.

For this reason, manual transfer of the structures became our protocol of choice to isolate the structure and rule out any interference from the substrate or the roots of the structures for the data presented in this manuscript. It seems reasonable that some of the applications of such an early-stage discovery will involve fundamental studies on isolated structures as well as applications on various substrates. Both are important and should not be dismissed. As shown by our literature survey in Table 1, manual transfer or use as-is on substrate is standard for high aspect ratio microstructures.

We found that the comment regarding the two-step process required an evidence-based context for the impact of our work to be reasonably assessed. We provide below a comparative table of conventional ways of obtaining high aspect ratio microstructures and their morphology (Table R1). In the context of what has been demonstrated experimentally, our process exceeds the performances of all methods in at least one aspect. First, it surpasses the aspect ratio of structures deemed ultra-high aspect ratio by a factor of 5. Next, to the best of our knowledge, it is the only process that is demonstrated on so many substrates, including polymers and textiles. Finally, the number of steps is low compared to lithography techniques, and the components required in our system are low-cost and low power.

Table R1 | Comparison of ultra-high aspect ratio structures and carbon structures reported in the literature

	Substrates used for process	Technique	Complexity of the shape	Transfer needed for applications	Number of steps	Ref.
Silicon (Si) ultra-high aspect ratio microstructures	Si	Lithography  • Au thin film deposition • Lithography (photoresist, exposure, pattern) • Chemical etch of Si 	Square micropillars Width: 20 μm Height: 200 μm Aspect ratio: 100	No transfer, patterned silicon substrate is used	6	[1]
Polymeric high-aspect ratio structures	Photopolymer adhesion layer deposited on a glass substrate	Femtosecond laser printing  • Liquid polymer is placed in a cell • Multi-photon lithography • Removal of uncured resin 	Continuous but not homogeneous morphology \varnothing : ~400 nm Length: ~ 200 μm Aspect ratio: up	Use as is Or Manual transfer	3 to 4	[2]

			to 500			
Ultra-high aspect ratio polymeric microstructures made by DLP 3D printing	Compatible DLP 3D printing substrate such as glass	 Digital Light Processing (DLP) 3D Printing Curing 	Morphology is segmented due to layer-by-layer \varnothing : ~20 μm Length: ~ 200 μm Aspect ratio: 100	Structures are attached to the base coat Use as is Or Manual separation and transfer	2 to 3	[3]
Ultra-high aspect ratio microstructures made by ultra deep X-ray lithography	Aluminum wafer	 Bake thick resist for several days Stack sublayers to obtain desired thickness Expose to X-ray lithography Post-exposure baking Development of the structures 	Microstructures Height: 7 mm Width: down to 18 μm Aspect ratio: up to 389	Manual separation and transfer	>5, with days-long steps	[4]
Glassy carbon microstructures	Silicon coated with a chromium adhesion layer	 Soft lithography: resin deposition, mold impregnation, curing Carbonization at 1100°C 	Various shape can be molded Height: ~10–20 μm Width: ~1 μm Aspect ratio: 10	Structures are attached to the substrate Can be manually transferred after carbonization	5	[5]
Micro carbon pillars	Stainless steel thin slice	 Laser-induced chemical vapor deposition (laser >0.5W) heat sample to gas decomposition conditions, use the laser to guide the growth 	Morphology is a continuous rod with smooth surface \varnothing : ~50–350 μm Length: < 3 mm Aspect ratio: <20	Manual transfer	2	[6]
Our carbon microstructure	Polymer, metal, quartz, glass, textile	 Transfer the catalyst layer on substrate Illuminate with laser (>532nm, >10mW) 	Morphology is a continuous rod with smooth surface \varnothing : ~2–10 μm Length: < 3mm Aspect ratio: up to 500	Use on substrate or Manual transfer	2	

We also accompany Table R1 with a summary of the morphology of the structures obtained with the methods listed in Figure R3.

[REDACTED]

Fig. R3 | Images of the microstructures obtained with the techniques described in Table R1, adapted from the respective publications, including silicon ultra-high aspect ratio microstructures [1], polymeric high-aspect ratio obtained by photolithography with a femtosecond laser [2], ultra-high aspect ratio polymeric microstructures made by DLP 3D printing [3], ultra-high aspect ratio microstructures made by ultra deep X-ray lithography [4], and glassy carbon microstructures [5].

Additionally, the lack of a defined target application for the printing technology is notable. While the authors discuss the technique in a broad sense, a specific application context would provide a more concrete framework for comparing this new technique to existing ones. By identifying and targeting a particular application, the authors could offer clearer insights into the potential advantages or limitations of their approach when contrasted with established methods, thereby enhancing the relevance and impact of their research within the field.

We have revised the manuscript to include the description of the potential applications and present the specific example of strain and temperature sensing in the manuscript. We'd like to point out a recent study by Kim et al. entitled "Multifunctional Intelligent Wearable Devices Using Logical Circuits of Monolithic Gold Nanowires" published in Advanced Materials [7], which highlights that a system with high sensitivity for strain and temperature changes is highly valuable for multifunctional wearable devices. In Kim et al.'s work [7], the nanowires had to be prepared in a polymeric film to obtain a deformable sensor (> 2 step process). The performance of the device is reported in their work as follows:

- 0.15–0.23% °C⁻¹ for the temperature sensor,
- a change of the strain sensor response reaching ~20% at 60° bending.

For the devices we demonstrated, we estimated the performance as follows:

- 0.44% °C⁻¹ for the temperature sensor in the 30–45°C range (relevant to the human body temperature) and higher sensitivity at higher temperatures,
- a change of the strain sensor response reaching up to 25% at ~60° bending.

In addition, we note that our device only requires a single carbon microrod and is fully recovered after bending for the few cycles we tested. We have highlighted these major advantages in the manuscript.

Reviewer #3 (Remarks to the Author):

In the revised version of the paper the authors have clarified all the queries raised by me. I am convinced that the present version is suitable for publication in Nature Communications

We thank reviewer 3 for their review of the work.

Reviewer #4 (Remarks to the Author):

The authors have provided substantial data towards answering my concerns about the paper and explored the potential applications of the unique carbon microstructures their printing technique is capable of generating. However, a few more changes need to be made before this work rises to the standards of significance commonly met by peer manuscripts published in Nature Communications.

First, my initial review asked for "a possible device... that leveraged the unique aspects of their system." While, indeed, the high aspect ratio of the generated structures may lead to superior performance for the structures generated in strain sensors, etc., I would like to see the authors directly test this

hypothesis by comparing against a common sensor material or other standard that does not benefit from the unique advantage of high aspect ratio conferred by the novel printing method presented by the authors. Most importantly, I would like to see data directly testing the hypothesis that the advantages conferred by the novel printing technique (i.e. high aspect ratio) are useful.

We have tested a common strain sensor for reference. As shown in Figure R4, we compared the response of our strain gauge sensor made of a single microrod with the response of a commercial strain gauge (SGD-4/120-LY43 - Omega). For this, the two devices were mounted on the same support for deformation, and the signal from each was measured simultaneously.

Our device shows change of $-59.9 \mu\text{V}/\mu\epsilon$ upon bending when the commercial gauge varied as $2 \mu\text{V}/\mu\epsilon$.

The high aspect ratio affords, in this case, a way to attain high degrees of deformation for the sensor. As described above, the deformation of the sensor is obtained with a single microrod without the need to embed it in a polymeric matrix as is commonly done with nanomaterials or microfibers [7-9]. The performance of the single microrod, for the strain of 0.025% experienced here, is notable as it is superior to gauge factors of 2-4 commonly reported for strain gauges [10]. This strain was limited by the flexibility of the assembled sensor stack. Flexing a mounted microrod without the commercial strain gauge in the stack allowed a strain of 0.55% to be applied without damaging the microrod. Furthermore, the rod can be deformed to form a full circle (strain >2%) as indicated in the previous response.

Fig. R4 | Strain gauge sensor performance of our single carbon microrod device. a, Schematics of the sensing setup. **b**, Response of the carbon microrod (red curve) compared to a standard strain gauge sensor (black curve) for an identical deformation.

Lastly, we note that the high aspect ratio of the carbon microstructure is not required for all applications. In fact, carbon growth in the plane of the catalyst layer could be sufficient to create a conductive path, as described in recent studies for flexible sensors [11].

Second, the performance of the fabricated carbon microstructures should strongly support the significance of the work and therefore should be included as part of the main text. Without this aspect of the story featured prominently, again, I feel that the demonstrated impact of this new printing strategy could be viewed as a less significant advance.

We have revised the manuscript accordingly.

1. Zhang, X., Yao, C., Niu, J., Li, H., and Xie, C., *Wafer-Scale Fabrication of Ultra-High Aspect Ratio, Microscale Silicon Structures with Smooth Sidewalls Using Metal Assisted Chemical Etching*. *Micromachines*, 2023. **14**(1): p. 179.
2. Cheng, H., Xia, C., Zhang, M., Kuebler, S.M., and Yu, X., *Fabrication of high-aspect-ratio structures using Bessel-beam-activated photopolymerization*. *Applied Optics*, 2019. **58**(13): p. D91-D97.
3. Tirado, M., Kundu, A., Tetard, L., and Rajaraman, S. *Digital Light Processing (DLP) 3D Printing of Millimeter-Scale High-Aspect Ratio (HAR) Structures Exceeding 100:1*. in *2020 IEEE 33rd International Conference on Micro Electro Mechanical Systems (MEMS)*. 2020.
4. Nazmov, V., Reznikova, E., Mohr, J., Schulz, J., and Voigt, A., *Development and characterization of ultra high aspect ratio microstructures made by ultra deep X-ray lithography*. *Journal of Materials Processing Technology*, 2015. **225**: p. 170-177.
5. Schueller, O.J.A., Brittain, S.T., and Whitesides, G.M., *Fabrication of glassy carbon microstructures by soft lithography*. *Sensors and Actuators A: Physical*, 1999. **72**(2): p. 125-139.
6. Zhou, J., Luo, Y.-s., Li, L.-j., Zhong, Q.-w., Li, X.-h., and Yin, S.-p., *Fabrication of micro carbon pillar by laser-induced chemical vapor deposition*. *Journal of Central South University of Technology*, 2008. **15**(1): p. 197-201.
7. Kim, T.Y., Hong, S.H., Jeong, S.H., Bae, H., Cheong, S., Choi, H., and Hahn, S.K., *Multifunctional Intelligent Wearable Devices Using Logical Circuits of Monolithic Gold Nanowires*. *Advanced Materials*, 2023. **35**(45): p. 2303401.
8. Huang, K., Xu, Q., Ying, Q., Gu, B., and Yuan, W., *Wireless strain sensing using carbon nanotube composite film*. *Composites Part B: Engineering*, 2023. **256**: p. 110650.
9. Duan, Q., Lan, B., and Lv, Y., *Highly Dispersed, Adhesive Carbon Nanotube Ink for Strain and Pressure Sensors*. *ACS Applied Materials & Interfaces*, 2022. **14**(1): p. 1973-1982.
10. Scholle, P. and Sinapius, M., *A Review on the Usage of Continuous Carbon Fibers for Piezoresistive Self Strain Sensing Fiber Reinforced Plastics*. *Journal of Composites Science*, 2021. **5**(4): p. 96.
11. Delacroix, S., Zieleniewska, A., Ferguson, A.J., Blackburn, J.L., Ronneberger, S., Loeffler, F.F., and Strauss, V., *Using Carbon Laser Patterning to Produce Flexible, Metal-Free Humidity Sensors*. *ACS Applied Electronic Materials*, 2020. **2**(12): p. 4146-4154.

REVIEWERS' COMMENTS

Reviewer #2 (Remarks to the Author):

The authors have comprehensively addressed all of my inquiries, and I agree with the publication of this paper.

Reviewer #4 (Remarks to the Author):

The authors have addressed all of my concerns. This manuscript is ready for publication. The microstructures are really quite well done.